# The polymorphism and tradition of funerary practices of medieval Turks in light of new findings from Tuva Republic

**Annie Chan**[1,2], **Timur Sadykov**[3], **Jegor Blochin**[3], **Irka Hajdas**[4], **Gino Caspari**[5,6]*

**1** Institute for Sinology, Ludwig Maximilian University of Munich, Munich, Germany, **2** Centre de Recherche sur les Civilisations de l'Asie Orientale (CRCAO), Paris, France, **3** Institute for the History of Material Culture, Russian Academy of Sciences, St. Petersburg, Russia, **4** Laboratory of Ion Beam Physics, ETH, Zurich, Switzerland, **5** Department of Archaeology, University of Sydney, Sydney, Australia, **6** Institute of Archaeological Sciences, University of Bern, Bern, Switzerland

* gino.caspari@iaw.unibe.ch

**Data Availability Statement:** All relevant data are contained within the paper.

**Funding:** The funders had no role in study design, data collection and analysis, decision to publish, or

## Abstract

The medieval Turks of the eastern Asian steppe are known for funerary finds exalting horsemanship and military heroism that thrived on intertribal warfare. Existing bodies of research on various categories of objects—which include architecture, stelae, grave goods and inhumations—are in depth but highly regionalized. As a result, our understanding of the archaeological culture of the Turks on a spatio-temporal scale commensurate with territorial shifts in their political dominion throughout the period of the Turk khaganates (mid-6th to mid-8th centuries CE) remains disjunct. The present paper addresses this problem of disparate data. We present a synthesis of the archaeological research of medieval Turks spanning Mongolia, southern Siberia, and Xinjiang in view of results of the excavation of medieval burials at Tunnug 1 in Tuva Republic—where Turkic remains are dispersed and not easily distinguishable from other funerary cultures of connecting time periods. We argue that Turkic funerary culture can be better characterized as polymorphic–the presence of different regional amalgams of burial traditions. The horse-and-human burials and commemorative *ogradka* known to be quintessentially Turkic are but one of the more dominant amalgams. This pattern of differential practices is congruent with the history of medieval Turks evolving as peoples of mixed lineages and political groupings, rather than people of a unitary culture.

## Introduction

Research delineating the purportedly distinctive material traits of Turkic funerary customs seldom makes reference to the enigmatic origins of the collective peoples to whom this pan-eastern steppe archaeological horizon may be ascribed. It is not a case of the lack of commensurate historiographical research, however, as inquiries into the ethnocultural composition of the Turks by way of etymology, linguistics, prosopography, and ethnology, make up a critical field of Inner Asian history in their own right [1–4]. The khaganate established by the Gokturks

preparation of the manuscript. The excavation project is conducted with the financial and logistical help of the Russian Geographical Society (N 53/04/2019) and financial support of Dr. F. Paulsen, the Society for the Exploration of EurAsia, and the Russian Ministry of Culture (project No. 656-01-1-41/12-18). Post-excavation works of T. S. and J. B. were carried out within the framework of the Programs of Fundamental Scientific Research of the Russian Academy of Sciences, State Assignments No. 0160-2020-0002 and 0184-2019-0011 respectively. G. C. was funded by the Swiss National Science Foundation, grant number P400PG_190982.

**Competing interests:** The authors have declared that no competing interests exist.

(552–603 CE) on grounds of their purported Turkic Ashina lineage, subsumed various Turkic and non-Turkic groups they subjugated across the latitudes from today's Mongolia and northern China to the Black Sea, among them the population under their former overlords, the Rouran. The inventory of artifacts that came to characterize Turkic archaeological culture took shape around the same time [5]. They include objects that belonged to the impedimenta of the martial lives of the cavalry and their steeds, and to commemorative sites used to perform ritual offerings and feasting. These objects represent the various practices of Turkic funerary tradition that are configured around different architectural elements—namely cairn burial, *ogradka* memorial enclosure, anthropomorphic stela, row of columnar *balbal* stone posts, royal epigraphic stela—that nonetheless vary greatly across Mongolia, northwest China, and southern Siberia.

This paper's inquiry emerged from an accompanying study of the medieval burial structures at Tunnug 1 in Tuva Republic, Russia, the heartland of the Turks in the second half of the first millennium. The meager material findings at the site prompted us to examine what constitutes "Turkic" burial culture through a cross-regional synthesis and comparison of reported "Turkic" funerary and commemorative structures. Given the stark regional variation in configuration of burial architecture and the differing prevalence of each structural type, outlining the geography of a "Turkic" archaeological horizon based on quintessential material traits that archaeological research has so far spotlighted may not be analytically productive. This paper argues that the political union that the Turk khaganates represented did not necessarily translate into homogeneity in burial form and practice; rather, "Turkic" burial culture should be characterized within the context of the *longue durée* of ethnogenesis and demographic movements that formed the eventual mixed populace assimilated by the Turk khaganates. Thus, to better assess changes in steppic funerary cultures on local and regional scales apropos of Turkic hegemony in the eastern steppe in the second half of the first millennium, it is necessary to shift our research focus from pronounced material traits to the polymorphism of funerary practices that have been characterized as Turkic.

## Research of medieval burials in Tuva Republic

Few medieval Turkic burials have been investigated in Tuva. There has been little new information since B. B. Ovchinnikova published the documentation of 21 sites in 1990 [6]; T. Sadykov has made a few additions and clarifications on the basis of Ovchinnikova's research [7: see map on p. 254]. The last monograph fully devoted to this period is *Древние тюрки в Центральной Туве* (*Ancient Turks in Central Tuva*), published in 2013 by the Institute for the History of Material Culture of the Russian Academy of Sciences [8]. It elaborates on the result of excavations in Central Tuva between 1965 and 1984. A large number of structures remain, however, unpublished, with the notable exception of burials with a horse [9]. Most of them have been discovered in Khemchik Basin and on the left bank of the Yenisei River in western and central Tuva. Prior to the Tunnug 1 excavations, only one burial with a horse was known in the Uyuk Depression. It was discovered during the excavations of the Arzhan 1 mound [10].

Compared to horse-accompanied burials, much more is known about cremations of the medieval period in Tuva. By 1995, more than 450 of them had already been excavated [11: 136] throughout the territory of Tuva. The number has since increased primarily as a result of rescue excavations. Nonetheless, most of these sites remain unpublished [12–15], and the data have not been synthesized since G.V. Dluzhnevskaya published *Pamyatniki yeniseyskikh kyrgyzov v Tuve (IX-XII vv.)* (Monuments of the Yenisei Kyrgyz in Tuva (IX-XII centuries)) in 1985. As the title of Dluzhnevskaya's publication suggests, all medieval cremations in Tuva

belong, most likely, to the Yenisei Kyrgyz, who invaded the territory of Tuva from the Minusinsk Basin and Khakassia, defeated the Uyghurs in 840 [16:6130, 17] and settled on the left bank of the Yenisei. Whether some of the cremations of the medieval period were part of the Turkic funerary culture, as the chronicles describe, has been questioned; the general consensus is that present archaeological evidence for Turkic cremation is tenuous, and equivocal at best.

## History and sites of research in southern Siberia, Mongolia, and Xinjiang

The Russian Altai is where field research of Turkic archaeological remains has been most prolific. The region is understandably the geographical focal point of interest as it not only has the highest concentration of archetypal Turkic burials, it is also where purportedly incipient Turkic structures dated to before the First Turk empire were discovered [18]. That the Altai might also have been the ancestral homeland of the Turks—specifically, "the political conglomerate of the Kok Turks" designated *Tujue* in Chinese sources that possessed "general cultural-ethnic uniformity" [19:264]—is also a popular theory deduced from Chinese biographical texts that chronicle geopolitical changes in Inner Asia since the disintegration of the Xiongnu. The name "Turk", however, did not appear in history until mid-sixth century when the Turks, then vassals of the Rouran, defeated the Turkic-speaking *Tiele* (*Gaoche*) and assimilated them [3: 25].

Given the lack of prosopographic details elucidating the nomenclatural conventions that aristocrat-led lineage federations followed and the volatility of group alliances [4, 20] in the intervening half-millennium between the Xiongnu and the Gokturks of fabled Xiongnu descent [3, 21], it can be safely assumed that the *Tujue* Turks constituted a "unified, but perhaps not uniform sphere" [19:264] and that the First Turk empire comprised ethnicities of Turkic and non-Turkic speaking groups [1:156].

As summarized below, the archaeological record across southern Siberia, Mongolia, and Xinjiang reveals a similar landscape of heterogeneity, one that is diverse at least in terms of mortuary ritual. The findings in these regions evince the presence of geographically segregated groups of varied linguistic, political and military lineages in a pluralistic empire whose growth propagated the burial customs ingrained in the deep cultural heritage of the Altaic steppe [cf. 18].

### The Altai and Minusinsk Basin

The Altai is undoubtedly one of the most studied territories occupied by the ancient Turks. The long history of archaeological research has generated a sizable body of literature on the subject, predominantly in Russian. Excavations of ancient Turkic structures in the Altai have been ongoing since V.V. Radlov excavated the Katanda site [22] as early as 1865. The monograph by A.A. Gavrilova [23] and *Tyurkskiye drevnosti Sayano-Altaya v VI-X vekakh* (*Turkic Antiquities of the Altai-Sayan region in sixth-tenth centuries*) by B. B. Ovchinnikova [6] became the most important compendium of Turkic materials published in the twentieth century. Subsequent field-defining studies include books [24, 25], as well as numerous books and articles by V.V. Gorbunov, G.V. Kubarev, V.D. Kubarev, Yu.S. Khudyakov, V.I. Soenov, N.N. Seregin, and A.A. Tishkin [e.g. 25–33].

The rich collection of objects recovered from the multi-period sites of Katanda in the Uyman steppe of the Russian Altai, comprising three cemeteries, has provided a comprehensive corpus of archetypal materials for identifying funerary remains of the Turkic period [34–36]. The collection features military equipment in iron, wood and bone (spearhead, sword, arrowhead, scabbard, quiver), horse riding gear in iron, bone and antler (stirrup, mouthpiece,

bridle and saddle hardware), battle attire in textile, leather, iron and bone (belt buckle, composite waist belt), and more uncommon finds including silver vessel with loop handle, Chinese silk damask and jewelry. The burial structures are in the form of low round barrows of 1.5–3 meters in diameter; roughly a third of them contain a horse and a human separated by a stone or wooden partition in the middle [36].

In contrast to Tuva, medieval burials with a horse outnumber cremations in the territory of the Altai. To date, more than 200 of them attributed to early medieval Turks have been discovered, along with 300 "memorial" Turkic enclosures, over 300 Turkic stone statues, and more than 90 runic inscriptions. By comparison, only up to 20 medieval cremations of the Kyrgyz have been investigated [32: 115].

Researchers have divided Turkic burials with a horse into several phases, although the nomenclature and criteria for division they apply slightly vary. The most developed scheme is perhaps the following: Kyzyl-Tash stage (second half of fifth centuries CE—first half 6 centuries CE); Kudyrge stage (second half sixth centuries CE—first half of seventh centuries CE); Katanda stage (second half seventh centuries CE -first half of eighth centuries CE); and Tuekta stage (second half of eighth centuries CE—first half of ninth centuries CE) [30].

In the adjacent Minusinsk Basin where the bulk of medieval monuments are attributed to the Yenisei Kyrgyz, Turkic burials appear to be in the minority. Researchers have dubbed them a special "Minusin" type, but they have not been well-studied. The question of when the ancient Türks first appeared in the Minusinsk Basin has thus not been resolved [31], but most researchers date this process to no earlier than the eighth century CE [29].

## Mongolia

The most emblematic assemblage of tombs and memorial complexes attributed to the Gokturks of the First Turk empire and the Second Turk empire is found in Mongolia. The strong artistic details of these remains–some of which indicative of the influence or presence of contemporary Chinese artisanship and funerary architectonic methods [37]–and the prodigious epigraphic record of Turkic military achievements eulogized in runic, Sogdian, and Chinese scripts are why field research of medieval archaeology in Mongolia has largely focused on these sites. The sites are concentrated in central and northern Mongolia, across Bulgan, Tuv, Övörkhangai, and Arkhangai aimags.

The memorial complexes are characterized by the presence of most, if not all, of the following structural components–a building of worship and an inscribed stele inside a graveyard, accompanied by an earthen burial mound, a series of *balbal* (upright commemorative columnar stone posts arranged in long rows that extend up to several hundred meters), anthropomorphic stone statues (362 are documented as of 2014 [38]), other earthen constructions such as low walls and moats. The most well-known epigraphic stelae are found at the sites of Bugut memorial complex (inscribed stele currently in Tsetserleg Museum (Bulgan sum), Kol Tigin (Hashaat sum), Bilge Kagan (Hashaat sum), Taryat (Taryat sum), Bilge Tonyukuk (Bain Tsokto, Nalaikh), Ongin (Övörkhangai aimag), Küli Çor (Ikhe-Khushotu), Ongut (Altanbulag sum) [38: 182–185, 39].

The singular royal tombs of the *shoroon bumbagar* form (translated as 'earthen rotundity' by Christopher Atwood in [37]) in Bayanuur sum (Bulgan aimag) and Zaamar sum (Töv aimag), dated to 7[th]-early 8[th] century ([37]; for optically stimulated luminescence (OSL) dates of terracotta and mortar samples from Bayannuur, see [40]), contain evidence of architectural and decorative murals, clay figurines, and coinage that reflects the percolation of Chinese burial rituals into Turkic funerary practice as the Tang Empire took increasing control over the heartland of the previous Eastern Turk empire.

Non-royal Turkic funerary architecture remains are, however, underexplored in Mongolia. These are structures better known in the Altai-Sayan region (Russia) including burial mounds and square enclosures known as ogradki. Seregin's [41] synthesis of the burial grounds of the early Medieval Turks in Mongolia notes that these sites are concentrated in the central and northern regions in Arkhangai, Bayankhongor, Bulgan, Selenge, Övörkhangai, Töv aimags. There is a scatter of sites in the western Bayan-Ölgii, Uvs and Khovd aimags. The easternmost find is the site of Togosiin Ovdgyin in Khentii aimag.

Results from several Russian-Mongolian expeditions in the past decade are beginning to fill this gap in understanding. An ogradka excavated at the site of Khuree Zuslan-I (Bayan-Ölgii aimag) with wood and charcoal samples yielding dates of 258–603 CE (95%) and 138–542 CE (95%) [42], and another one at Hushuun Denzh-04 (Arkhangai aimag) [43] provide important evidence of the development of Turkic memorial structures in central and western Mongolia in the early medieval period up to the demise of the First Turk empire.

Burial mounds have also been studied. A human-and-horse burial furnished with iron weaponry and silver belt and bridle ornaments comparable to typical Turkic burial finds from the Russian Altai (e.g. at Balyk-Sook [44]) was excavated in Shanaga, Bukhmurun sum, Uvs aimag in 2011 [45]. Another human-horse joint burial was excavated at Syrgal-II in Bayan-Ölgii aimag [46: 29]. At another site in the same aimag, Kubarev [47] reported a joint male and female burial (Mound 8) and a likely a male cenotaph (Mound 10) dated to 652 CE (*terminus post quem*), and a series of stone circles at the cemetery of Khar-Yamaatyn-Gol [47].

The progress made in field research, however, does not yet address the patchiness of the study of Turkic burial complexes in Mongolia. Seregin [41: 92] contends that while current known site distribution may be reflective of past nomadic patterns of movement and settlement, it is more likely an artifact of the varied extents of field investigations in different parts of the country.

## Xinjiang

In Xinjiang, the majority of sites where burial remains characteristic of the Turkic funerary tradition have been discovered are in Bogda (Bogeda) Mountains, the northern ranges of east Tian Shan (Fig 1). The burial structures are predominantly in the form of surface round earthen-stone cairns over a vertical earthen pit, in which there is often a side chamber.

Multiple human-horse burials were discovered at the site of Xigou Cemetery in present-day Fukang 阜康 during excavations in 2010 [48]. M2, 4, 10, 16 are side-chambered structures housing dual inhumations of a horse and a human; the latter is placed inside the side chamber. The head of the deceased human is almost always in the west whereas the direction of the horse head may be east- or westward. The side chamber is more commonly found along the northern wall, along the N-S axis, although it also occurs on the western wall, such as at Ningjiahe Reservoir [49] and Qirentuohai [50] cemeteries. There also exist burials with two sets of horse remains on two separate tiers, and a human in the side chamber of the lower tier; examples include M16 of Xigou Cemetery [48] and M32 of Gangou Cemetery [51]. This double-tiered grave structure is also found at a few other sites, including Qirentuohai Cemetery [50] (XIA 2004) where it housed one set of horse remains and one set of human remains respectively on upper and lower tiers. There is a case of burial with two sets of human remains (an adult male and a child) along with a horse on the upper tier in M66 at Ningjiahe Reservoir Cemetery [49]. In all cases, the deceased human assumed the supine position.

Vertical earthen pits without side chambers have also been attributed to the medieval period but most of these graves contain few diagnostic objects. It is difficult to estimate how many of the structures are in fact Turkic burials given that descriptions provided in the

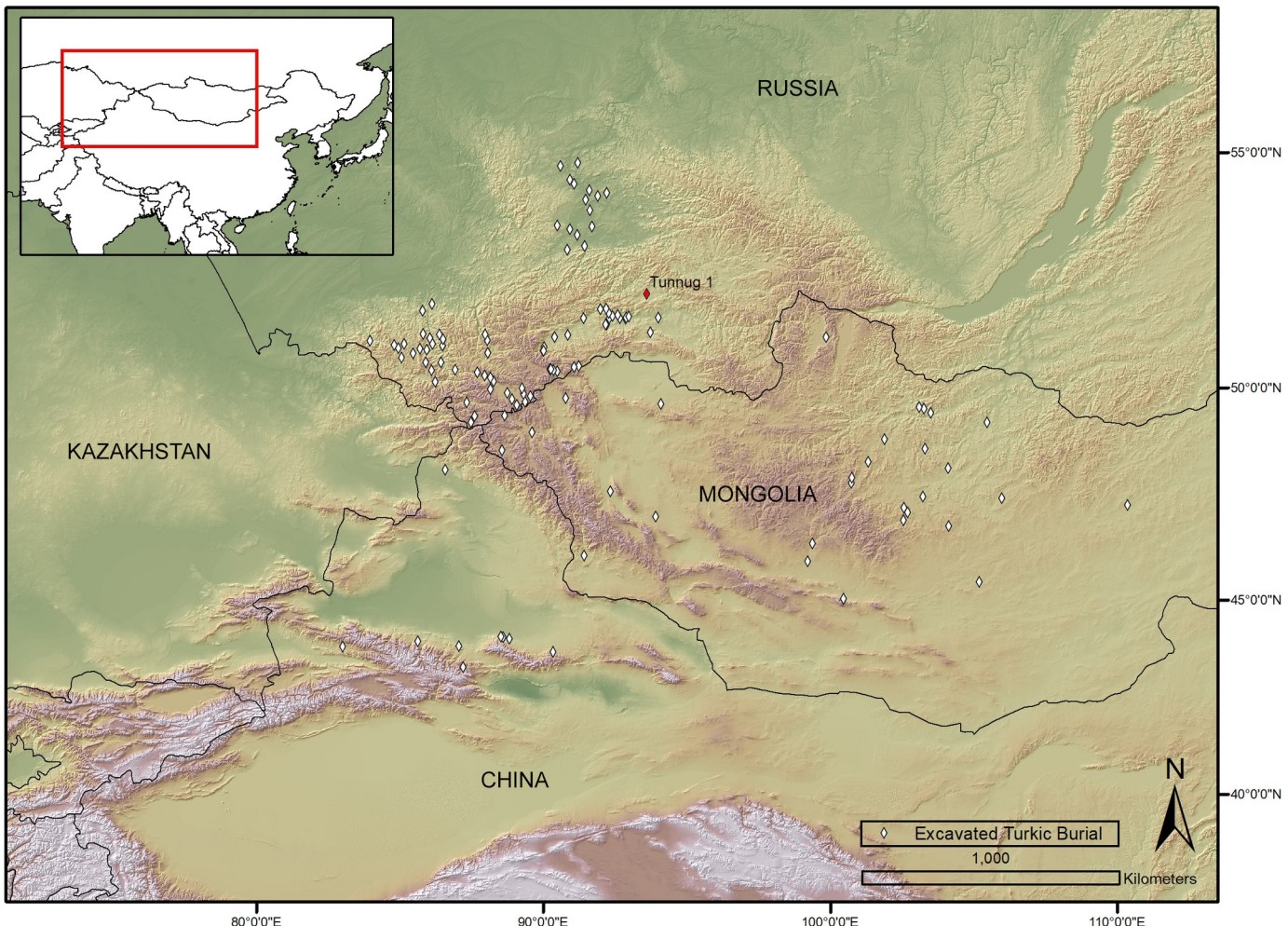

**Fig 1. Distribution of Turkic burial structures in Inner Asia (adapted from [55] for sites in Altai-Sayan region and Mongolia.** Map created by G. Caspari using ArcMap 10.4 by ESRI. We acknowledge the use of SRTM 1 Arc-Second Global Digital Elevation Model).

Chinese excavation reports are oftentimes cursory. The identification is made more difficult due to the fact that this subterranean burial form combined with a surface cairn has been associated with a wide range of material cultures and time periods, most commonly the Iron Age (second half of the first millennium BCE).

Burial objects found inside these graves are mainly bronze implements (buckle, mirror, ornament), iron weaponry (most commonly, arrowhead, dagger), tack (iron stirrup, bit), and bone bow grips. At Xigou Cemetery, a plethora of bronze, iron, and bone objects as well as a small collection of silver, stone and ceramic objects were found buried with the deceased. Bronzes include a wide range of *diexie* (蹀躞)-girdle ornaments with engraved designs, buckles, fragment of a mirror. Iron objects are mostly arrowheads and stirrups. This combination of accoutrements is also present at the site of Gangou in Mulei County in M32 and M47 although the human and horse remains are aligned north-south, with the horse's head in the north in both burials and the human's one in the north and one south [51]. Similar engraved heart-shaped and owl-faced (梟面形) bronze girdle ornaments and buckles have been found at Ningjiahe Reservoir Cemetery [49] and Ergonghe Cemetery [52] both excavated in 2011.

Bone bow grips (弓弣) have been found in M17 and M33 at Ergonghe Cemetery [52], Ningjiahe Reservoir Cemetery [49], and Gangou [53].

Some of these structures have been scientifically dated, including M25 (1395±30 BP; cal. 640–720 CE), M36 (1170±30 BP; cal. 860–1000 CE), and M37 (1155±25 BP; cal. 880–990 CE) at the site of Baiyanghe Cemetery (XIA 2012), M32 (1340±60 BP; cal. 580–830 CE (94.1%)) and M43 (2250±25; cal. 310–200 BCE (61.6%); sample contamination due to structural damage likely explains the skewed dates) at Gangou (XIA 2015a: 20–21), and M37 (1155±25 BP; cal. 880–990 CE) at Ergonghe Cemetery (XIA 2015b). Dating in few other contexts was possible due to the presence of Tang dynasty *kaiyuantongbao* coin, such as in the case of the vertical side chamber graves (洞室墓) of M12 and M13 at Baiyanghe Cemetery [54] or mirror of the Tang dynasty designs at Xigou Cemetery [48] and Gangou [53].

## Results of the excavation at Tunnug 1

Field research was conducted under licenses No. 0434–2018 and No. 0590–2019 issued to T.S. from the Institute for the History of Material Culture, Russian Academy of Sciences. The permits were issued by the Russian Ministry of Culture. In current archaeological research, the Uyuk Valley in Tuva Republic, where Tunnug 1 is located, is known primarily for its large burial mounds of the Early Iron Age [56]. The river terraces of the valley are dotted with burial mounds in long, roughly east-west aligning rows. International scholarly literature of Tuva's archaeological heritage has mostly been focused on the emergence of the "Scythians" in the eastern Eurasian steppes, their relation to monumental tombs, and the hierarchical organization of nomadic societies. However, archaeological remains dating to after the Scythian period are also frequently found during rescue excavations. A preliminary survey conducted in 2017 in the southern zone of the Uyuk Valley where monumental burial mounds are markedly scarcer than on the northern river terrace led to the establishment of the Tunnug 1 excavation project with the aim of uncovering one of the earliest and largest monumental tombs of the Early Iron Age steppe [57].

The initial excavations were carried out on the southern periphery of the main mound where a mix of Scythian period, Kokel and Turkic period stone structures were first detected through a geophysical survey [58]. Most of these structures are clearly separable from each other and represent consistent chronological units (Fig 2). The exception is an amorphous Kokel cemetery, consisting of numerous burial structures built over an extended period of time without recognizable pre-planning patterns. The Kokel cemetery superimposes on a chain of Scythian stone circles fringing the main burial mound [59]. The general form of the separable peripheral structures is a low-lying cairn overlaying a perimetric stone circle with a central pit. Adjacent to these larger cairns are satellite structures in the form of small (0.5-3m) stone circles. Excavations have uncovered human inhumations, cremations, animal bones, an assortment of metal horse tack, weaponry, and ornaments, an anthropomorphic stone stela, and scattered ceramic fragments, which are typical finds of Turkic burials.

While there are layers of frozen soil underneath the main Early Iron Age burial mound, the structures in the southern periphery are situated within an active frost layer. Over the course of the year, the condition of the archaeological remains oscillates between frozen and thawed. The soil frequently cracks open allowing the upper carbon rich layers to fill the gaps. This results in a criss crossing pattern of cracks in most of the planigraphy (Fig 3A). Freezing soil lifts parts of the layers up leading to the emergence of a wave-like pattern across much of the periphery (Fig 3B).

During the summer months, the periphery thaws completely. The site is situated within the floodplain of a river. High groundwater is a frequent issue when excavating the lower layers of

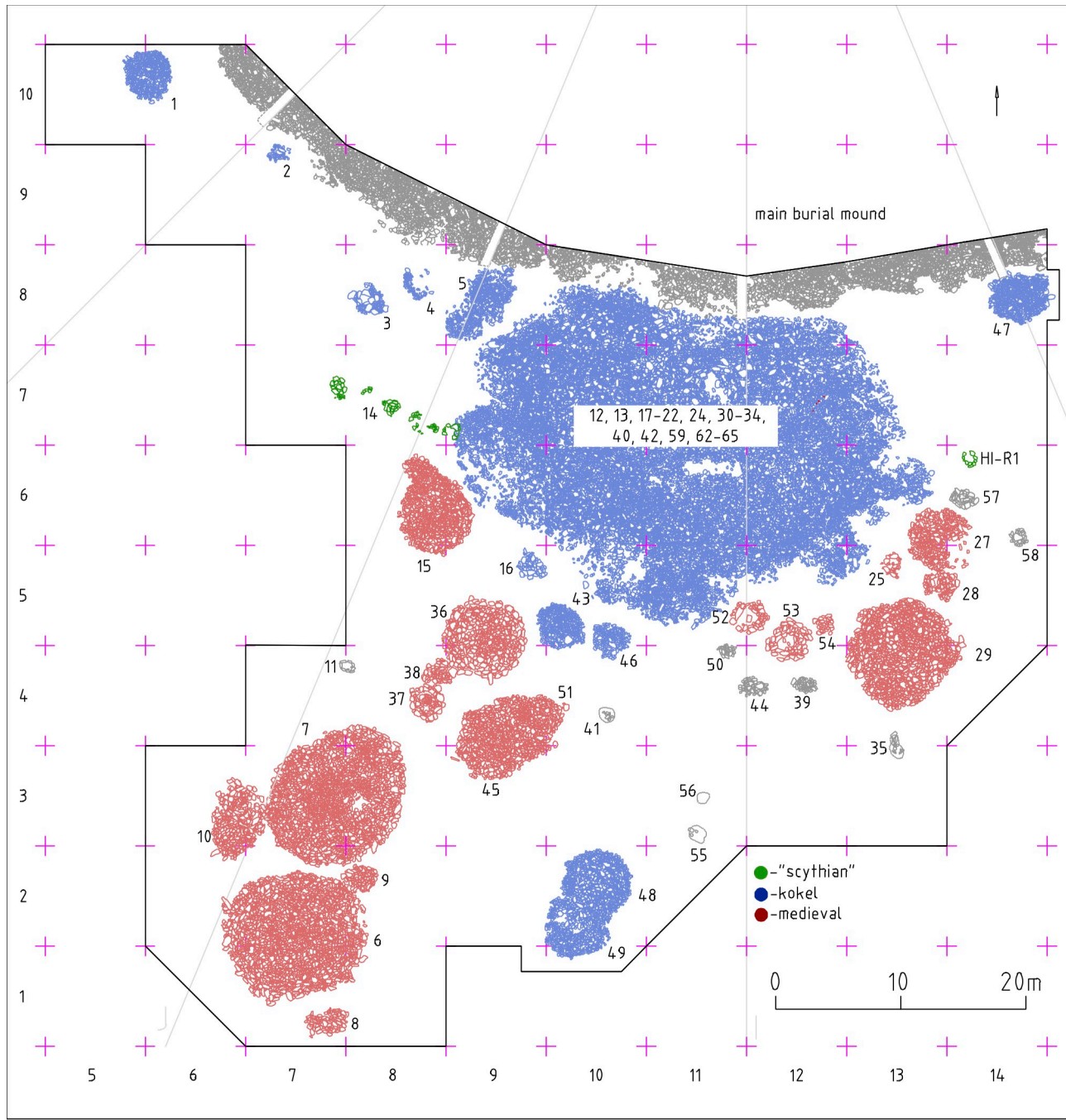

**Fig 2. The southern periphery of the Tunnug 1 site including a number of medieval stone monuments (red) adjacent to a large amorphous Kokel cemetery.**

a structure, in particular the burial pits. During early spring, layers are easily separable. With the onset of high waters in June and July, the soil becomes waterlogged and visual and haptic separability of layers is reduced substantially. Gravity-inclination induced solifluction is limited since the site lies in a flat area. The freeze-thaw cycle and associated movements of the soil, however, can lead to displacements of artifacts and bones. Even in cases where a structure

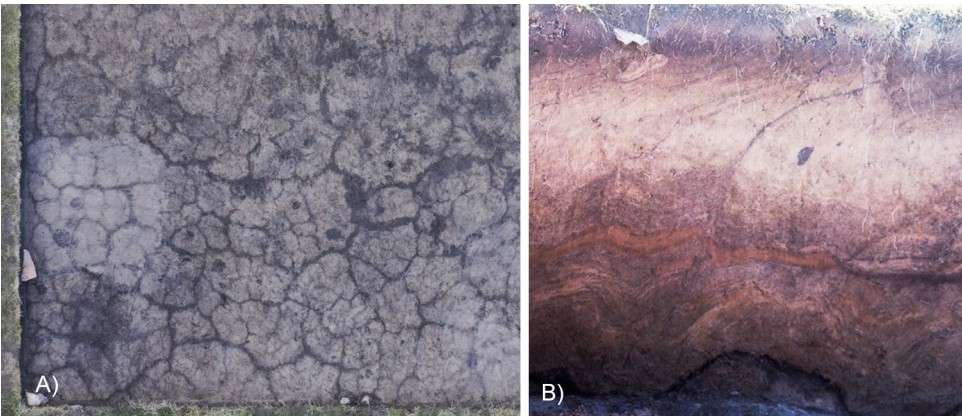

**Fig 3. Effects of freeze thaw cycles on the excavation site.** A) Criss crossing cracks in the planigraphy. B) Wave patterns in the stratigraphy.

has clearly not been subject to anthropogenic disturbance, parts of the skeleton and inventory can be missing. A limited number of structures in the periphery have been impacted either by looting or other destructive anthropogenic activities like the intentional destruction of the multiple burial at the center of the Kokel cemetery [59, 60].

The first excavation campaign took place in 2018. It revealed that Tunnug 1 was used for funerary ritual purposes over a period of more than two millennia [61]. Although no other large Early Iron Age monuments could be detected in the flood plain of Uyuk River [62], geo-physical analysis of the periphery of the main burial mound including geoelectric, geomag-netic, and ground penetrating radar surveys showed a plethora of smaller stone monuments to its south that were not visible on the surface [58]. The dominating archaeological feature of the southern periphery is an amorphous accumulation of burial features associated with the Kokel culture [59]. The burial ground contains burial pits with single and multiple burials covered by stones sourced from the Early Iron Age burial mound. Some features represent so-called "over-vessel-mounds"- which are stone mounds built on top of one or multiple buried vessels which may or may not contain human remains. Later secondary interments were found between stone layers of the amorphous structure. 87 individuals dating to the second and fourth century CE, many of them showing chop marks, slice marks, penetrating lesions, and blunt traumas were discovered [60, 63].

To the south of the archaeological features associated with the Kokel culture, a number of medieval structures were found. In accordance with the requirements of Russian archaeolog-ical authorities, each individual "object"—a discrete structure—receives its own stratigraphic profile and a plan of each contained archaeological layer [64]. Most of the medieval structures are located in clusters comprising a central structure 6, 7, 15, 36, 45, 29, around which smaller structures are aggregated (Table 1). The central structures in the clusters feature both horse-

**Table 1. 14C dates of bone samples.**

| Material | Lab Code | Funerary Structure | 14C age (BP) | C% | C:N | Calendar age (2σ) |
|---|---|---|---|---|---|---|
| human bone | ETH-108832 | 7 (sk. 16, main burial) | 1254 ± 22 | 42.6 | 2.9 | CE 674–870 |
| human bone | ETH-108834 | 45 (sk. 73, main burial) | 1303 ± 22 | 43.4 | 2.8 | CE 661–775 |
| human bone | ETH-108837 | 38 (sk. 63, child burial structure appended to structure 36) | 1305 ± 22 | 43.8 | 2.8 | CE 660–775 |
| human bone | ETH-108838 | 6 (sk. 102, main burial) | 1328 ± 22 | 43.5 | 2.8 | CE 653–774 |
| human bone | ETH-108841 | 29 (sk. 52, additional child burial) | 1306 ± 23 | 45.1 | 2.9 | CE 660–775 |

and-human inhumations (6, 7, 45) and possibly cremations (15, 29, 36). In the accompanying structures (or at the edges of the central structures), burials of children, dogs, and animal bones are found, but there are no adult burials.

The finds uncovered can be grouped into several categories: items equipped for mounted combat comprising tack (iron horse bits (Fig 4), stirrups (Fig 5), fasteners (Fig 6)), armament (iron arrow points, bone arrow whistles), apparel (bone buckle, belt ornaments), ceramic vessel fragments, a Tang dynasty *kaiyuan tongbao* 開元通寶 coin, and an anthropomorphic stela. The construction and material finds of each structure are detailed below.

## Structures 6–10

Structures 6 and 7 are two mounds with pits at the center. Both structures have been heavily disturbed by a combination of post-depositional human activities and natural weathering processes. A horse skeleton (further described below) and human bones were discovered in the central pits of both structures. Dog bones were found in the southeastern part of structure 6. In the northern part of structure 7, sheep and children's bones were found.

Scattered in the central pits were iron arrowheads, bone arrow whistles, and various items of tack, which include bone and bronze belt buckles, bridle ornaments, bone fasteners, and stirrups (Figs 4–8) Fig 7, Fig 8.

The horse in structure 6 is probably a male as four canines were found. The skeleton consists of fragments of a cranium and mandible, fragments of vertebrae, ribs, fragments of the shoulder, radial, whole anterior metapodia, first and second phalanges of the forelimbs, hoof phalanges of the forelimbs, fragments of the pelvis, diaphragms, the femur, the right tibia, the right talus, the first phalanges of the hind limbs, and many fragments of the postcranial skeleton. There are right and left PM2 teeth with characteristic traces of bridling with metal bits. Judging by the degree of wear on the incisors, the animal was about 15 years old.

An entire skeleton of an old horse (male) was also found in structure 7. The skeleton consists of fragments of a cranium and mandible, fragments of vertebrae, fragments of ribs, a fragment of the right shoulder, a fragment of the right ulnar, first and second phalanges of the forelimbs, hoof phalanges of the forelimbs, a fragment of the pelvis, a right left fragment of the femoral head, right and the left calcaneus, posterior metapodia, the first phalanx of the hind limbs, and many fragments of the postcranial skeleton. Judging by the wear pattern on the teeth, the animal was about 20 years of age. One tooth PM2 was found but traces of bridling with metal bits were not present.

The form of the surrounding structures 8–10, are difficult to discern due to the soil conditions and mixed or partially disturbed contexts. Ash deposit in 8, 9, and 10 is hardly discernible as a result of waterlogged soil. The pit in structure 9 is faint and difficult to make out; only very few animal bones were found here. Structure 10 has two filled-in stone circles with three pits. In pit 2 (in the southern part of the northern circle) scattered children's bones were identified. Horse bits and stirrup (Figs 4 and 5) were found in pit 3 in the southern stone circle.

## Structure 15

Structure 15 is a cairn constituted of a stone circle, 4.5m in diameter, with an inner ring outlining a central pit, a smaller stone circle to the northwest and a surface layer of filled-in stones (Fig 9). No artifacts were found in the central pit; a ceramic fragment of coarse temper with etched lines was found in the mound. No direct evidence of burial or cremation has been found here, but similar kurgans are widespread in Tuva, where a very small number of calcined bones are found on the surface or in small pits under a mound. Due to the soil

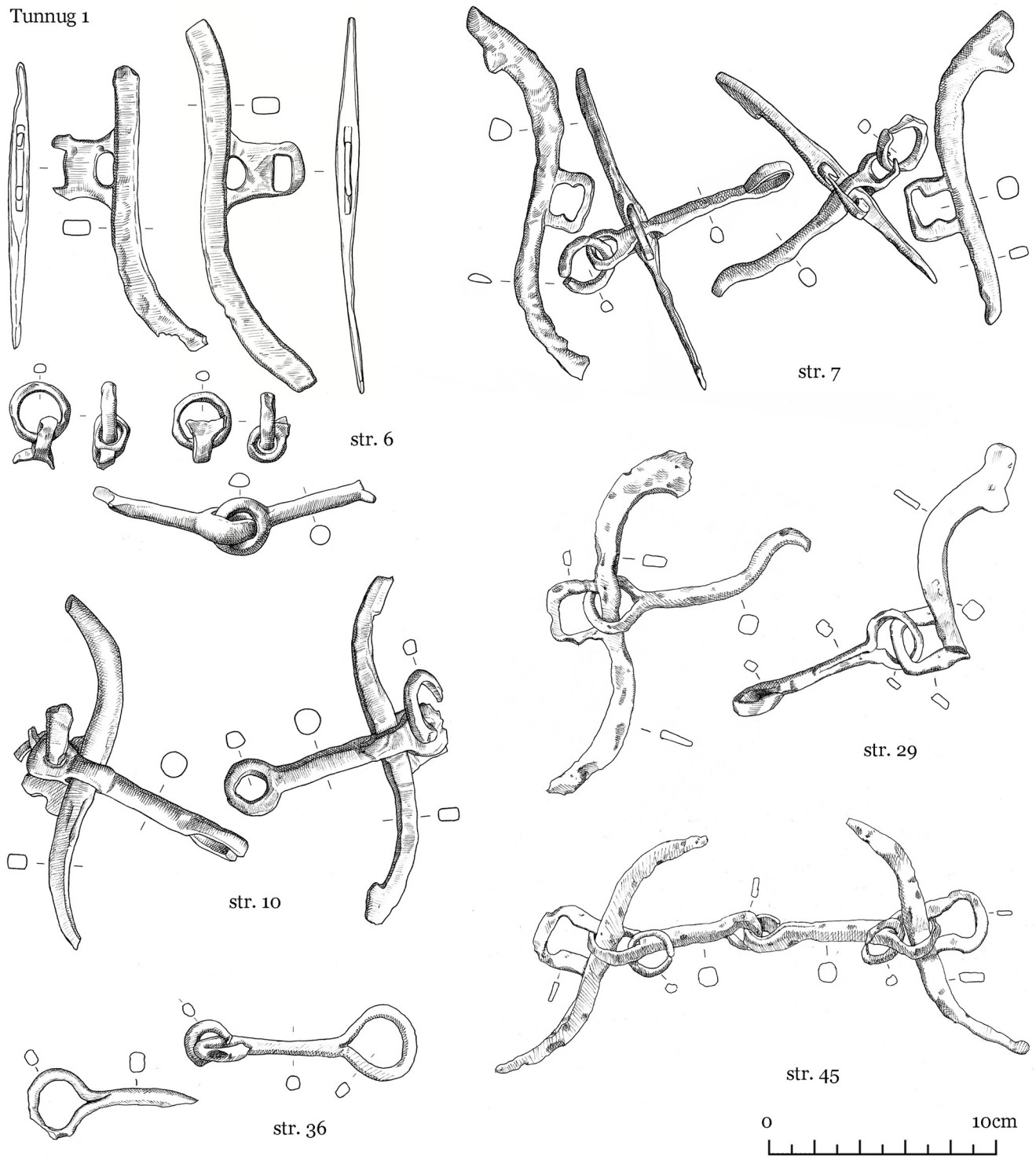

**Fig 4. Tunnug 1.** Medieval artifacts. Iron horse bits.

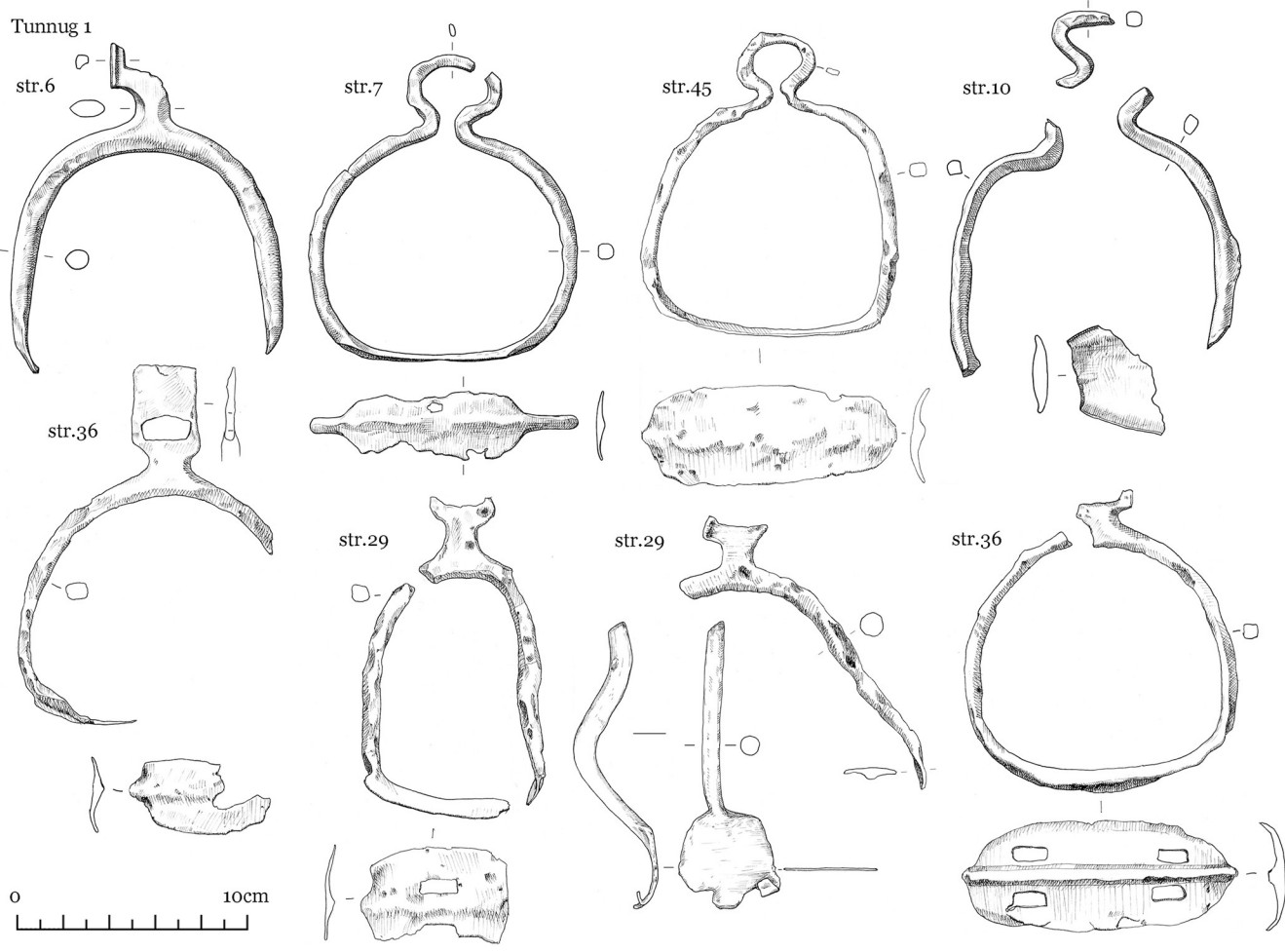

**Fig 5. Tunnug 1.** Medieval artifacts. Iron stirrups.

conditions, the calcined bones may often not be visible [65]. An additional pit contains a child burial with a Tang dynasty *kaiyuantongbao* coin [61: fig 6:1].

### Structures 36–38

Structure 36 consists of a ring of big stones with an overlay of smaller stones. An accumulation of iron artifacts was found near the center which included bridles, stirrups, buckles, and arrowheads (Figs 4–7). No direct evidence of burial or cremation was found.

Structures 37 and 38 are additional structures connected to the southwestern edge of structure 36 (Fig 10) that are similar in form and content. Bone remains of a child inhumation and caprid bones were found in both structures (Fig 11).

### Structures 45 and 51

Structure 45 has a horse-and-human burial (Fig 12) contained in a pit under a central cairn enclosed by a perimetric circle. The remains of the horse and the human are both oriented east-west but in opposite directions. The horse's head is oriented to the west, the human cranium was displaced and is found next to the horse skull. There is a burial of a 8–9 year old

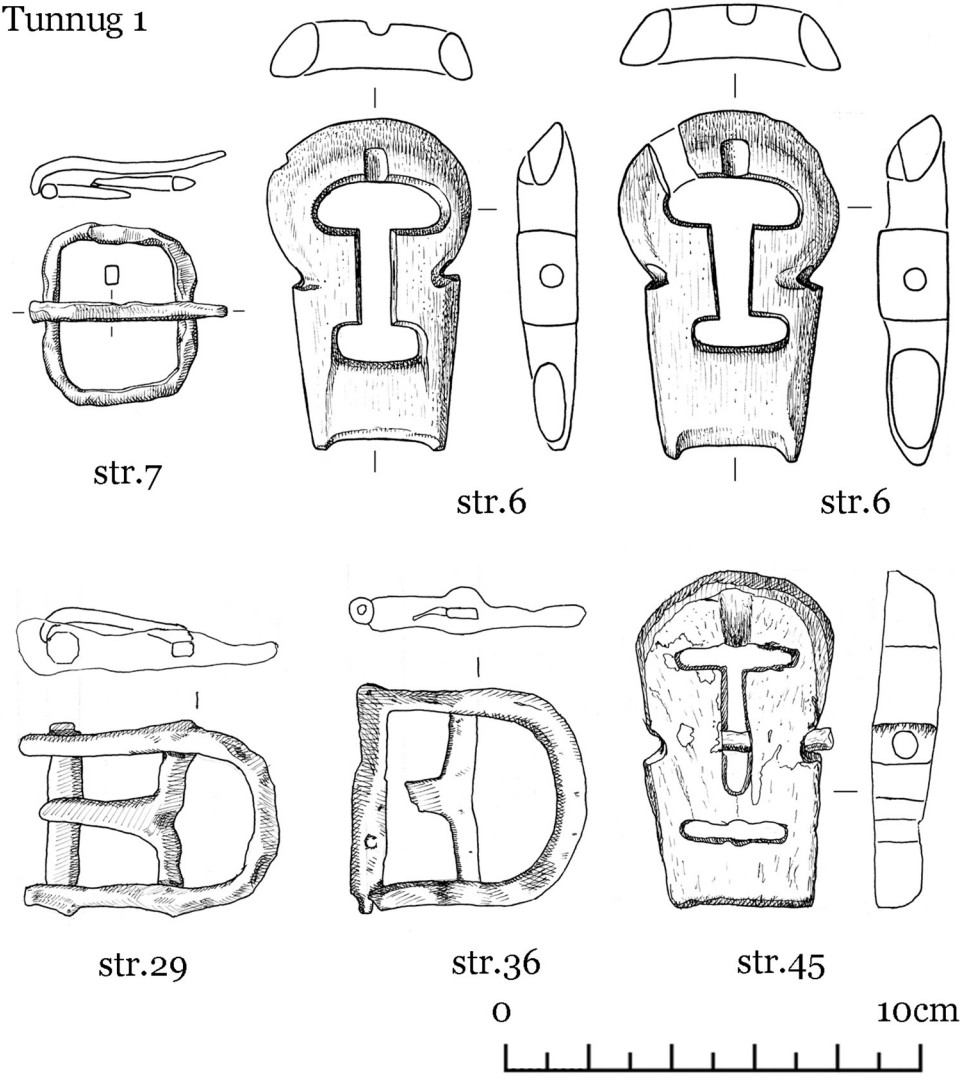

**Fig 6. Tunnug 1.** Medieval artifacts. Iron and bone fasteners.

individual (skeleton 73) at the center. The inventory (Figs 4–8) includes one iron stirrup, bridles, a knife, 3 arrowheads, bone buckles and bone grips.

The horse discovered in structure 45 is a yearling. Judging by the configuration of the dentition on the preserved lower jaw and by the degree of accretion of the epiphyses of the tubular bones, the age of the animal is determined to be approximately two years. The horse is probably a female as there are no signs of canine teeth on the lower jaw. The height of the horse at the withers along the length of the anterior metapodium, which was fully formed following V. O. Witt [66], can be determined at about 135 cm, that is, according to his classification, the size of a horse is below average. Based on the length of the metapodia and the width of its diaphysis, the degree of thin-leggedness of an individual according to Brauner [67] is 13.7%, that is, the horse is rather thin-legged. L.L. Gaiduchenko [68] classifies horses with such size and gait characteristics as low grade "riding horses". Thus, it can be assumed that even though the horse was not high-grade, it was still quite suitable for riding.

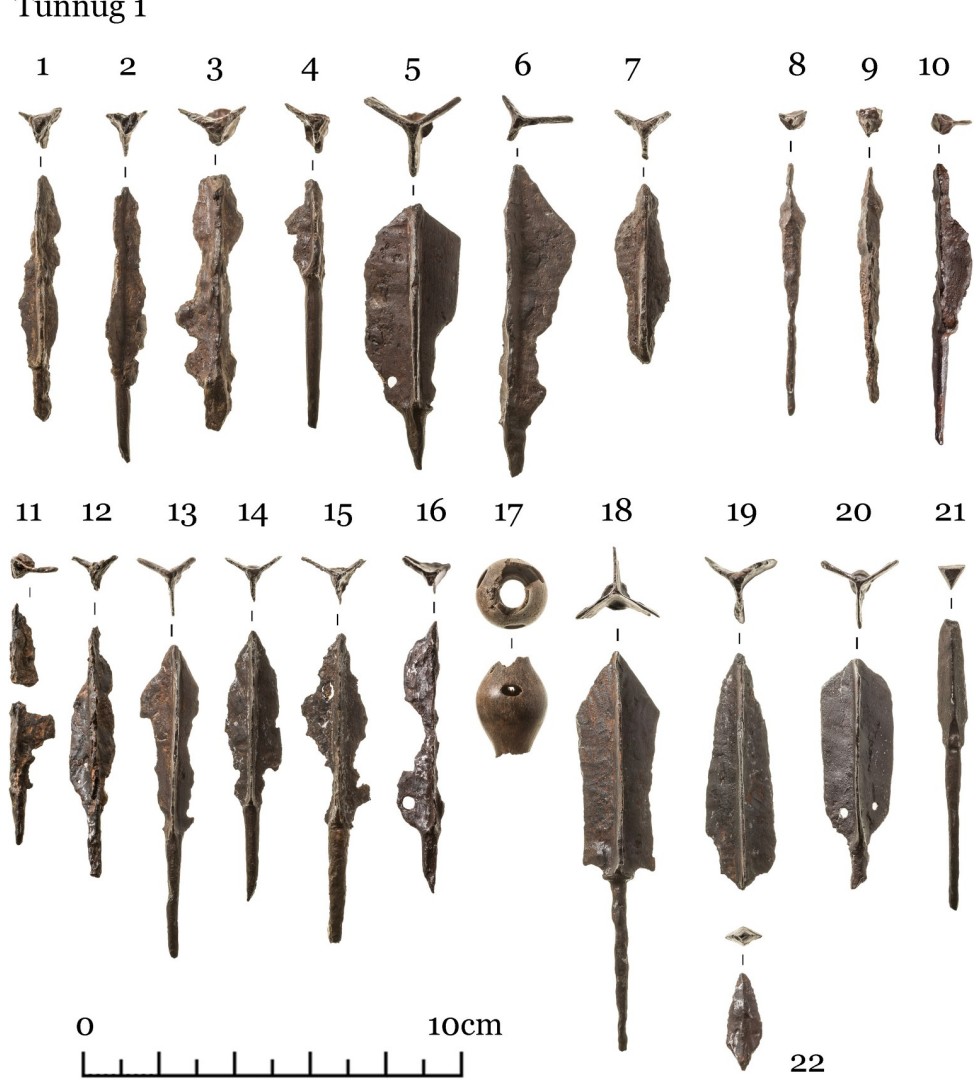

**Fig 7. Tunnug 1.** Medieval artifacts. Arrowheads.

No traces of bridging were found on the front lower teeth, which are still deciduous. Since the horse is quite young, it most likely did not die of natural causes but was sacrificed for some kind of ritual and cult purposes. The animal's skull is heavily fragmented but it has not been preserved in its entirety. How the animal was killed is not clear. No traces of trauma or penetrating wounds were found on the bones of the postcranial skeleton. A rounded hole was found, about a centimeter in diameter with uneven edges, on one of the last thoracic vertebrae, at the base of the spinous process on the left side. The hole did not go through. It is unclear where it originated but its position and depth indicate that it was certainly not a fatal wound.

Skull fragments and vertebrae of a roe deer skull were also found in structure 45. The animal is a female, since the antler bases are clearly absent on the fragments of the skull. Some dog bones were discovered; they include a fragment of the scapula and the lower part of the right hind paw (tibia, calcaneal, ram, dorsal metapodia). They belong to a medium-sized dog that stood around 40 cm tall and weighed 10–15 kg fully grown.

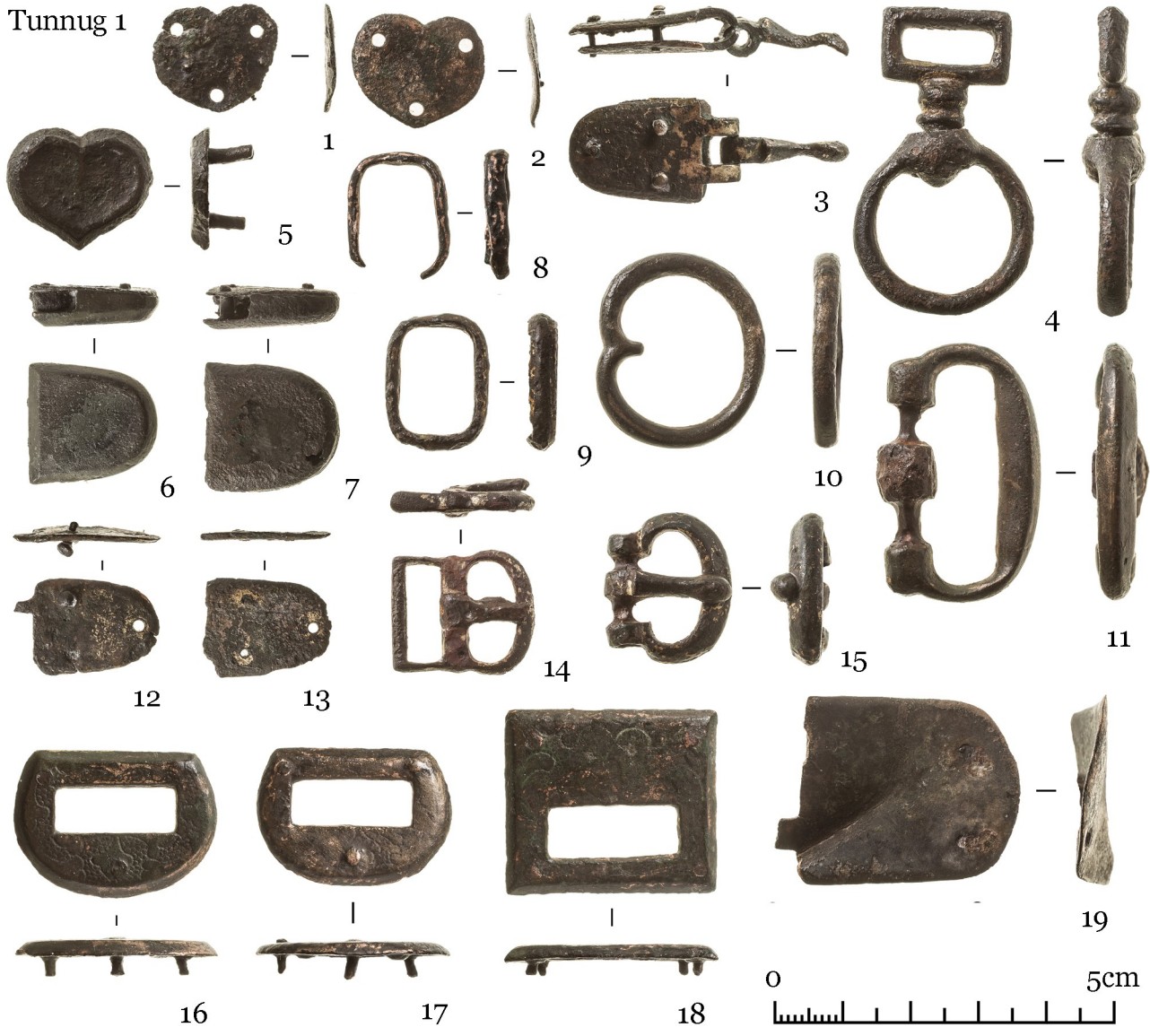

**Fig 8. Bronze finds from structures 6 and 7.**

Structure 51 is an additional structure on the northeastern edge of 45 with three stone encircled pits (51–2, 51–2, and 51–3). A fetal skeleton (skeleton 80) and dog bones were found in its vicinity.

## Structures 29, 52–54

Structure 29 is a stone mound (Fig 13) consisting of a perimetric stone circle and an inner ring outlining a central pit. Iron items (Figs 4–7) and a ceramic vessel were found in the central pit. There are also several fragmented human, dog, and sheep bones in the pit. An additional dog burial was found in the western part of the kurgan and two child burials were excavated in the southern and eastern part of the kurgan. An anthropomorphic stela (Fig 14) was discovered on the edge of the mound. One end of the stela shows faint traces of a carved face typical of the corporeal style of Turkic anthropomorphic stelae.

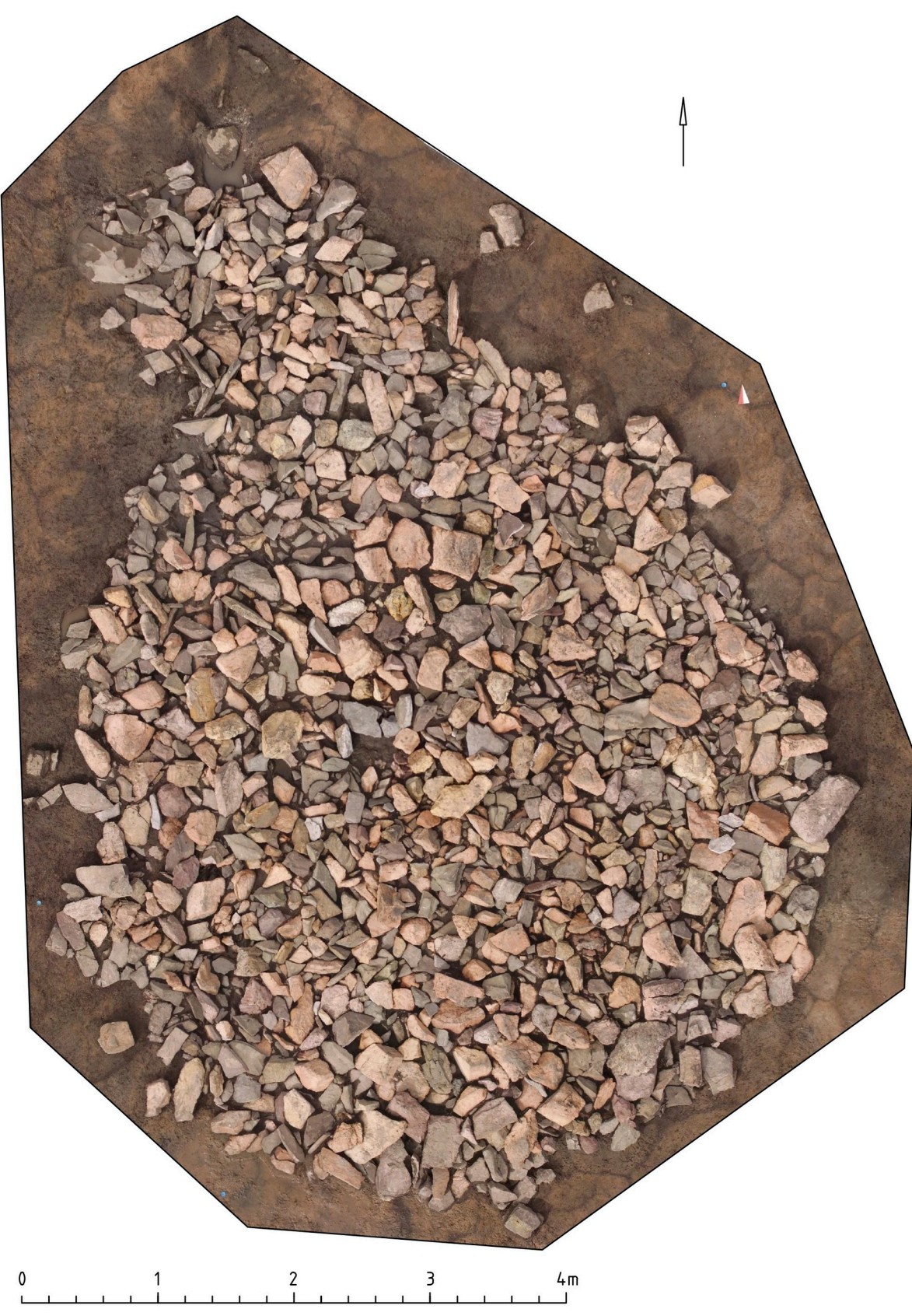

**Fig 9. Tunnug 1.** Orthophotograph of structure 15 before removing the stones covering the circle.

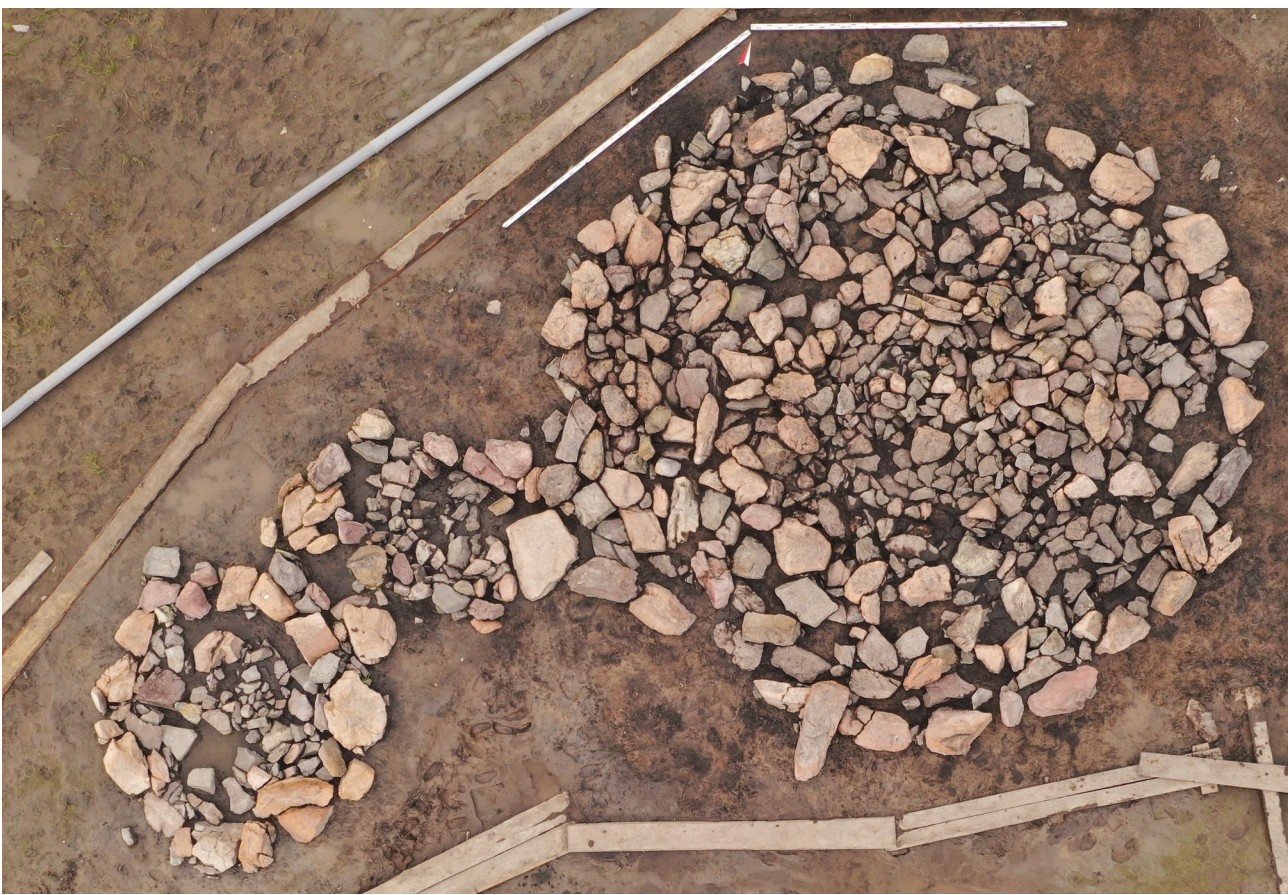

**Fig 10. Tunnug 1.** Structures 36–38.

52–54 are three adjoining stone circles that may also be auxiliary structures to Structure 29. Only four ceramic fragments were found in 53. The pit of structure 54 contained calcinated bones.

### Structures 25, 27 and 28

Structures 25, 27, and 28, located north of structure 29, are possibly not associated with the medieval period, and may be attributed to the next phase of occupation at the site. Structure 25 contained a deposition of a complete horse harness with numerous bronze decorations under a small stone paving. Structure 27 is a stone kurgan of 4 m in diameter. A millstone was found at the north-eastern edge of the mound. Below it, there was an amorphous pit with a cremation. Silver belt ornaments, an iron arrowhead, and a part of a bronze mirror were found inside the pit. Structure 28 is another stone circle enclosed pit. It only contains unidentifiable iron and ceramic fragments. The stratigraphic relationship between 27 and 28 is unclear.

## Discussion

### Structural types and site heritage

Structures dated to the medieval period at Tunnug 1, including those attributed to Turkic burial culture, are found south of those of Early Iron Age (9th century BCE) and Kokel (2nd - 4th century CE) [59, 69]. The distribution of later period burial mounds around a central

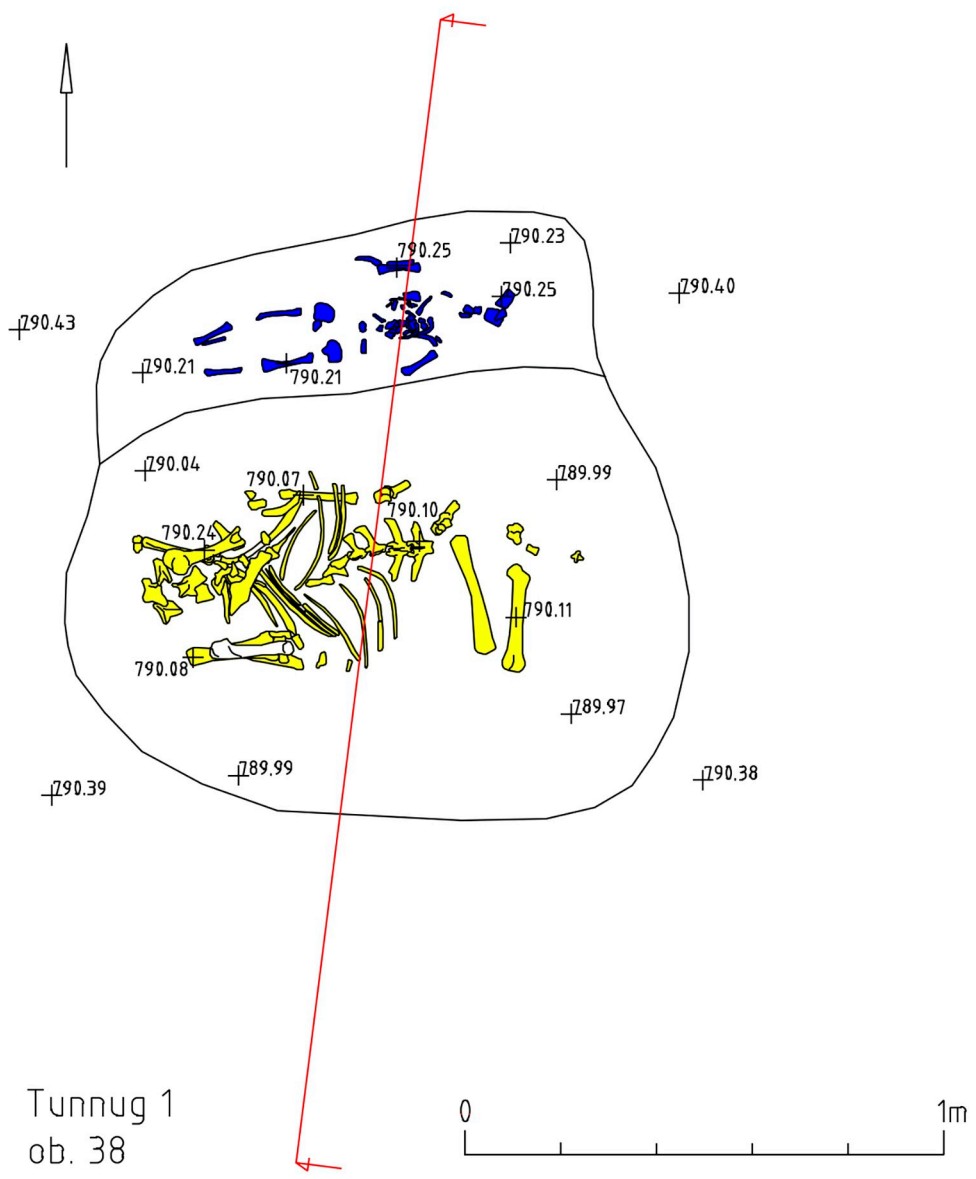

**Fig 11. Tunnug 1.** Structure 38.

kurgan dated to the early Iron Age shows that Tunnug 1 was a nucleus of funerary and commemorative activities for an extended period of time. The layout appears to be consistent with Kubarev's [27] observation that the Turkic burials are often grouped around earlier kurgans with the exception of a configuration of two kurgans and two enclosures, which occurs individually. The traditional placement of Turkic burials east of earlier structures [27] is not present at Tunnug 1, but the preference for orienting burials eastward [1, 70] is noted (see discussion in next section). In Tuva Republic, human with horse burials are most commonly found with human head to the east and horse head to the west.

Results of studies of Turkic burials in Mongolia are limited, thus little is known about the spatial organization of Turkic period structures relative to earlier ones. There is, however, a growing understanding of the chronology and configuration of *ogradki* (square memorial

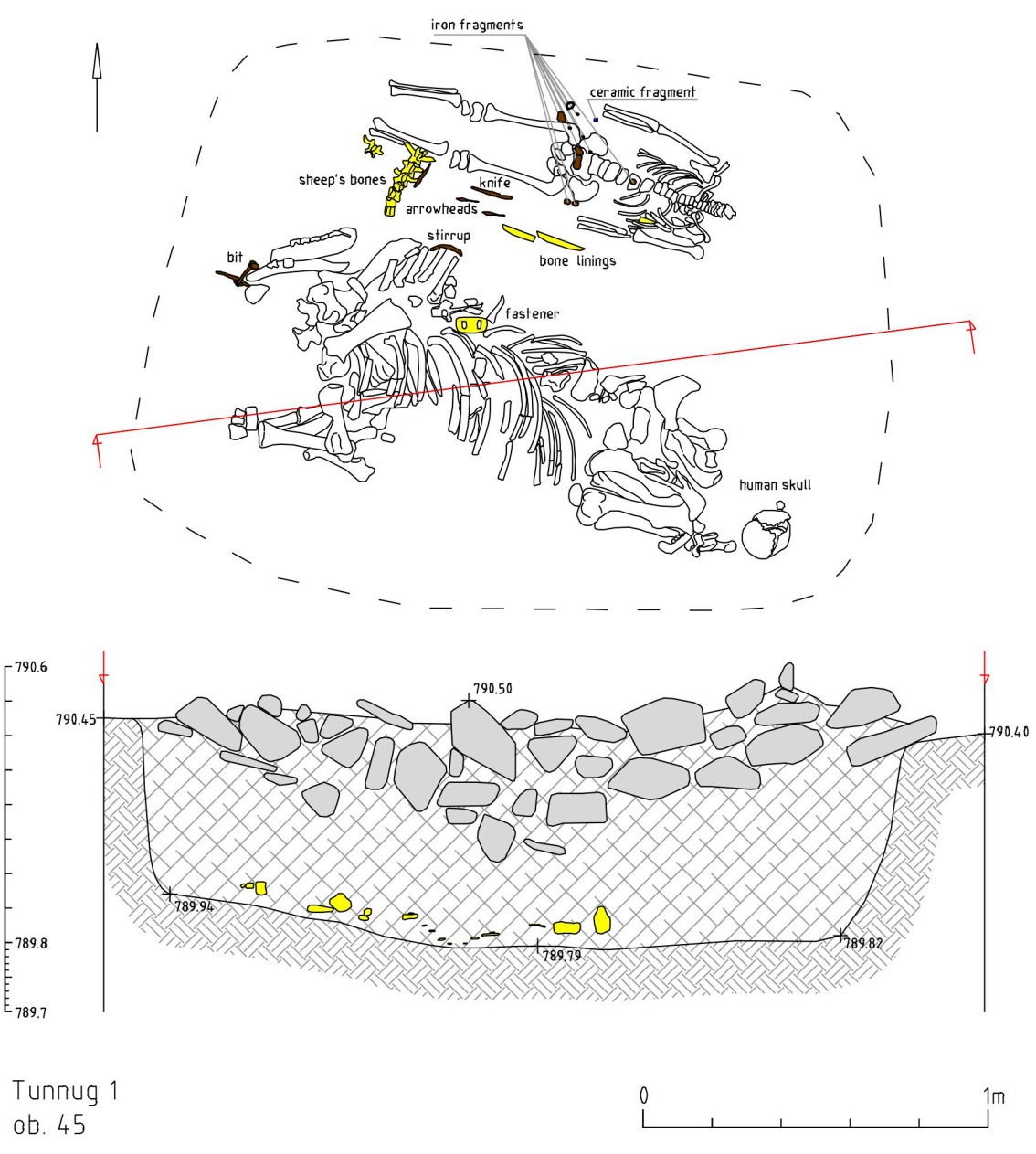

**Fig 12. Tunnug 1.** Structure 45.

enclosures) in western Mongolia owing to fieldwork conducted by a number of Russian-Mongolian teams. Besides these structural remains, memorial complexes located predominantly in northern and central parts of the country showcase the various building elements of Turkic funerary architecture and the spatial association between the subterranean funerary structures (burial mounds, ogradki) and the accompanying surface commemorative monuments (*balbals*, anthropomorphic stelae, epigraphic stelae). Hayashi [71] proposes viewing the *ogradka*, the *balbals*, and the anthropomorphic stela as a "minimum [structural] unit" that can be scaled up by incorporating more monuments.

The Turkic burials discovered in the eastern Tian Shan region of Xinjiang have a surface construction of a low round earthen-stone mound. In the rare cases where the site plan is

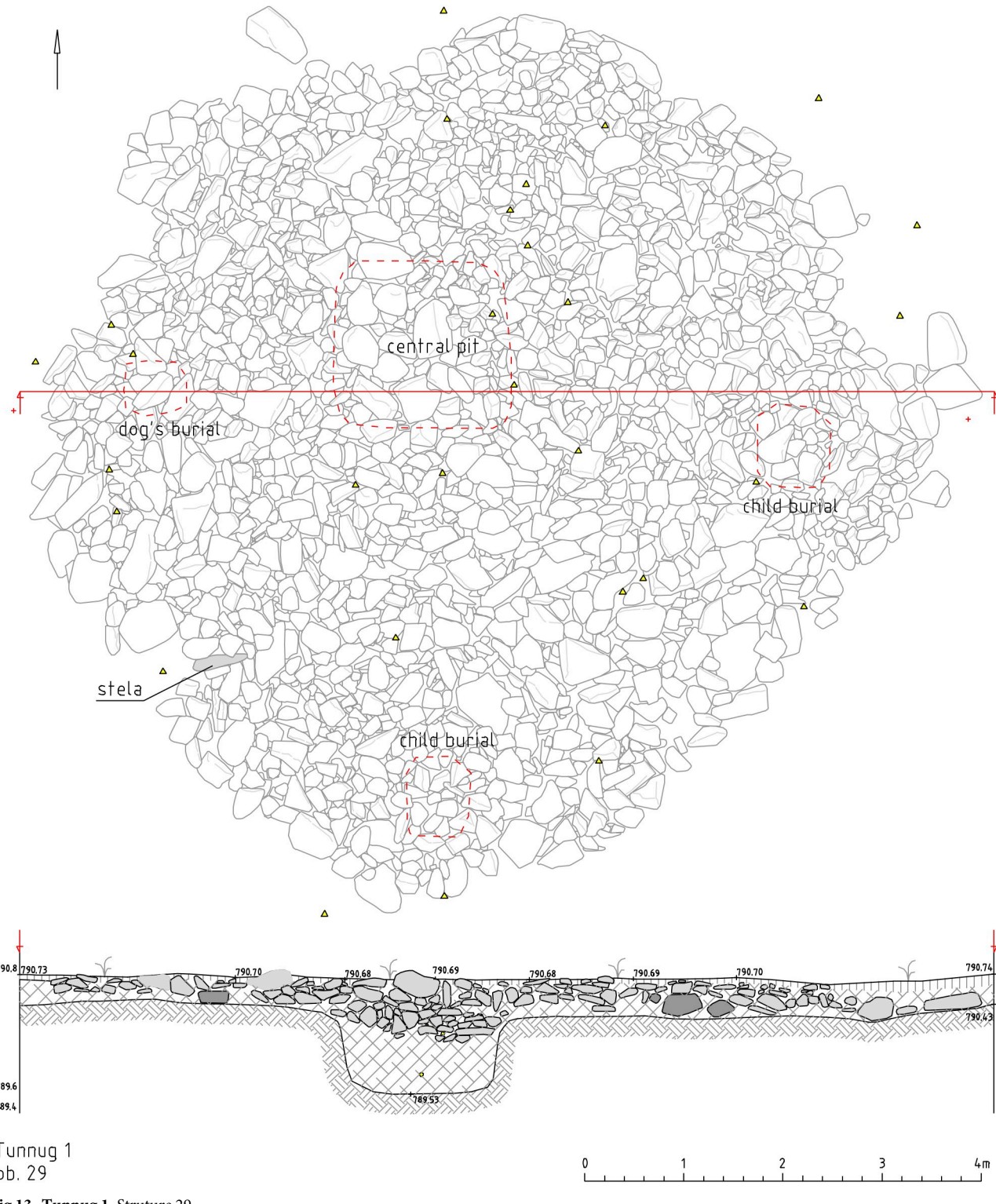

**Fig 13. Tunnug 1.** Struture 29.

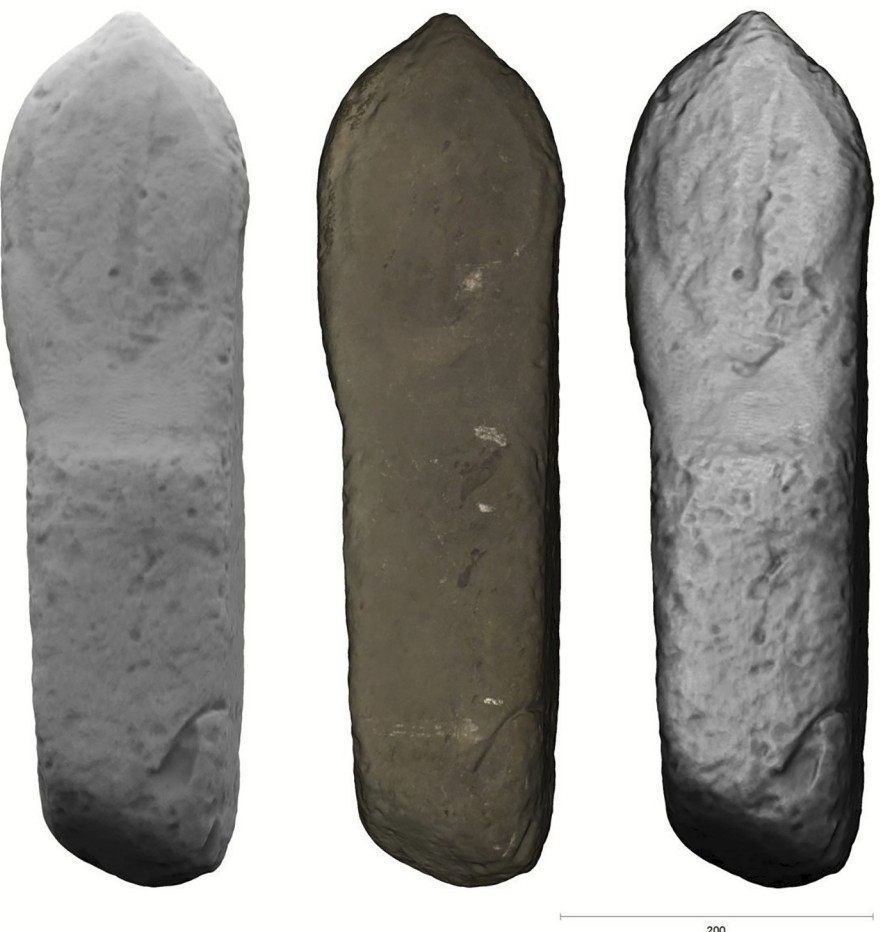

**Fig 14. Tunnug 1.** Turkic stela found at the edge of structure 29: standard model (left), textured (middle), contrast enhanced (right).

provided in the excavation report, e.g. for Ergonghe Cemetery [52] and Xigou Cemetery [48], the Turkic structures are shown to be intermixed with structures of Iron and Bronze Age periods as well as contemporaneous non-Turkic structures. Balbals and anthropomorphic stelae are rare in the eastern Tian Shan (a few stelae have been discovered in the vicinity of counties of Jimsar and Mulei in Changji Hui Autonomous Prefecture); anthropomorphic stelae are most concentrated in western Tian Shan and the Chinese Altai, in the prefectures of Ili, Bortala, Tacheng, and Altai.

## Types of interment and subterranean structural layout

Structure 45 at Tunnug 1 contains primary inhumations of horses and humans oriented east-west but facing opposite directions. Due to anthropogenic disturbance, the weathering effects of freeze-thaw cycles, and solifluction, the morphology of the burial pits of structures 6 and 7 is unclear but there were horse and human bones as well.

Joint human-animal burials are a distinct feature of Turkic burials in the Altai, numbering 15, compared to only one from Tuva (Ulug-Horum) 73: 196. It has been considered to be a local phenomenon of the Altai and possibly a form of deviation from ceremonial norms 73: 197, but the function of human-horse burial in Turkic funerary custom remains subject to debate.

Another notable feature of joint human-horse burials that appears to vary regionally is the partition between the human and the horse remains. This spatial divider could be in the form of vertical stones such as at the site of Justyd XII, wooden trunk such as at Džolin I, [27: 28]. There are also cases where the deceased is placed inside a wooden or stone lined coffin [27: 25]. This structural partition is also found inside horse-human burials in Xinjiang where the deceased is placed in a side chamber at the bottom of a vertical pit. Excavators have also documented several two-tiered burial arrangements such as at Xigou Cemetery where two sets of horse remains accompanying a human burial at the lower level are found.

Another interment feature in question is the presence of cremation. In the conglomerate of structure 15, 29, and 36–38, the central structures were possibly made for cremation (or functioned as a cenotaph), and the inhumations, which are child burials, are located in the auxiliary stone circles and not in the central pits. Evidence of cremation is equivocal at Tunnug 1 due mostly to solifluction-induced disturbances to the strata in the structures 15, 29, 36. There are, however, more distinct traces of cremation in structure 27 and around structure 54.

The *ogradka*, a distinctively Turkic funerary structure and linked to cremation in historical texts, is not found at Tunnug 1. Most Russian scholars consider the *ogradka* a cenotaph. The *ogradka*—generally translated as "memorial/mortuary enclosure" in Russian literature—is a quadrangular, low slab-fenced structure occasionally marked by a stone or wooden post at the center adjacent to a stone recess that have been found to contain tack, weaponry (specifically, spearheads), and armory, the second and third categories of which are seldom found in interment burials [72]. The discovery of these personal battle impedimenta predominantly in the *ogradki*, and deposited in a "miniature" receptacle no less, lends weight to the argument that the objects were used as proxy for the dead following the rituals for cenotaphs [72, 73]. It is suggested that the *ogradka* serves as a dwelling place for the spirit of the deceased [74: 88] and that the beginning of this commemorative practice dates back to before the First Turk empire. The Russian Altai has the highest concentration of *ogradka*. Ogradki found in Mongolia share the same structural characteristics and contain similar objects of interment as those discovered in the Altai e.g, [26, 43].

The "cenotaph" hypothesis remains controversial since observed variations in the structural composition of the ogradka, the attribute and placement of perimeter slabs, corner stones and connecting memorial structures (balbals, anthropomorphic stele), as well as the types of burial objects (or the absence thereof) make it difficult to generalize the nature of the ogradka. The debate relates also to the description of the interment process in Chinese historical texts. Two aspects are in question—first, whether the structure described can be identified as what is known archaeologically as *ogradka*; and second, whether it functioned as a cenotaph in lieu of burial or was a structure used for cremation.

The "cremation" debate concerns the manner in which the corpse is burnt and the ashes are interred. The *Book of Sui* 隋書 (*Suishu* 84: 1864) and the *Book of Zhou* 周書 (*Zhoushu* 50: 910), respective official records of the reigns of Western Wei and Northern Zhou (535–557 CE; 557–581 CE), and of Sui dynasty (581–618 CE), redacted retrospectively in early Tang, provide almost synchronized descriptions of the cremation ritual (differing descriptions in the latter source bracketed henceforth). The deceased (and their possessions) is placed and burnt on a horse on a chosen day; their ashes are then collected to be interred (await the time of internment). (Those who died in the spring or summer await the grass and trees to yellow and the leaves to fall whereas those who died in the autumn and winter wait for the foliage to be in full bloom before their ashes are interred in a pit.) A wooden (stone) post is installed to mark the grave.

That cremation was a characteristic interment practice of the Turks as described in the Chinese sources has been largely refuted by archaeologists. The reason being there are, so far,

simply no cremation burials that can be unequivocally identified as Turkic. Cremations are widespread in Tuva and are interpreted as Kyrgyz burials from the 9th c. CE, but this tradition could have appeared in Tuva even earlier. Descriptions of the Kyrgyz peoples in the *Institutional History of Tang* (*Tang Hui Yao* 唐會要) corroborates the practice of cremation among the Kyrgyz [75: 1784](Wang 1935: 1784 (vol. 100)). The data from Tunnug, as they stand, do not contribute directly to the cenotaph/cremation debate for the Turks.

A number of hypotheses have been put forward to explain the discrepancy between archaeology and text, which include a shift in burial custom, and the interment location and timeframe. First, a change in burial custom early in the 7th century CE, as described by a passage in *The New Book of Tang* (*Xin Tangshu* 新唐書) recorded in 628 CE describing the Turks opting to inhume the dead in grave mounds in defiance of their cremation tradition [19, 71] would explain the absence of cremation among Turkic burials, the majority of which are dated to the Second empire. Second, the ashes were absent or difficult to trace because they might not have been scattered inside the ogradka or there is a half year gap between cremation and interment, as the chronicles recount [18, 71].

Scholars who rejected both the "cremation" and "cenotaph" hypotheses have maintained that these arguments are untenable given current archaeological evidence; they thought it more viable to consider the *ogradka* a designated memorial place where various forms of rituals were performed and ceremonial offerings to spirits and deities were made, thus explaining the presence of ashes and burnt bones in some of these enclosures (see [28] for a synthesis).

## Burial objects

The most distinctive characteristic of Turkic burials across Inner Asia, is its repertoire of burial objects. Russian archaeologists such as V.V. Gorbunov, G.V. Kubarev, V.D. Kubarev, N.N. Seregin, A.A. Tishkin, have published compendiums of these artifacts based on results of their fieldwork in the Altai-Sayan region [e.g. 25–30, 76] but comparative analysis across the wider region is still lacking in this respect. G.V. Kubarev's *Alttürkische Gräber des Altaj* [27] has a comprehensive summary of finds in the Altai in English and German.

The artifacts yielded from the burials of the medieval period at Tunnug 1 includes metal objects: iron arrowheads, iron stirrups, bronze buckles, bridle ornaments, horse bits, a bronze mirror, a Tang *kaiyuan tongbao* coin; bone objects: bow grips, buckles, arrow whistles; ceramic fragments; and an anthropomorphic stela. This collection is consistent with the usual preponderance of ornamented horse tack, combat apparel, and weaponry and the occasional occurrence of items of Chinese origin in Turkic burials. Balyk-Sook Cemetery in Altai Republic has one of the most representative collections of cavalry combat gear. Kurgan 11 is a rare noble kurgan where a large amount of tack, armor and weapons was found attributed to a deceased male and four horses. The presence of these thoroughbreds and the use of silver for various accessories—e.g. horse bridle set adorned with silver and rare gems—suggest the high social status of the deceased [44: 70]. The cheekpieces, bits, and stirrups are morphologically consistent with what has been discovered in burials across Mongolia [77, 78]. This high degree of uniformity in funeral rites [76:144] appears to go hand in hand with a heraldic trend in military metalworks [18: 11] throughout the region under the political dominion of the Turks.

Burials in Xinjiang have yielded similar repertoires of bronze, iron, and bone objects. Most notable are the heart-shaped and T-shaped bronze bridle ornaments (Fig 15), which are present at Xigou Cemetery, Ningjiahe Cemetery and Ergonghe Cemetery. In addition, silver and ceramic objects were discovered. At Xigou Cemetery, for example, silver ornaments—earrings and plague—were found in secondary inhumations [48].

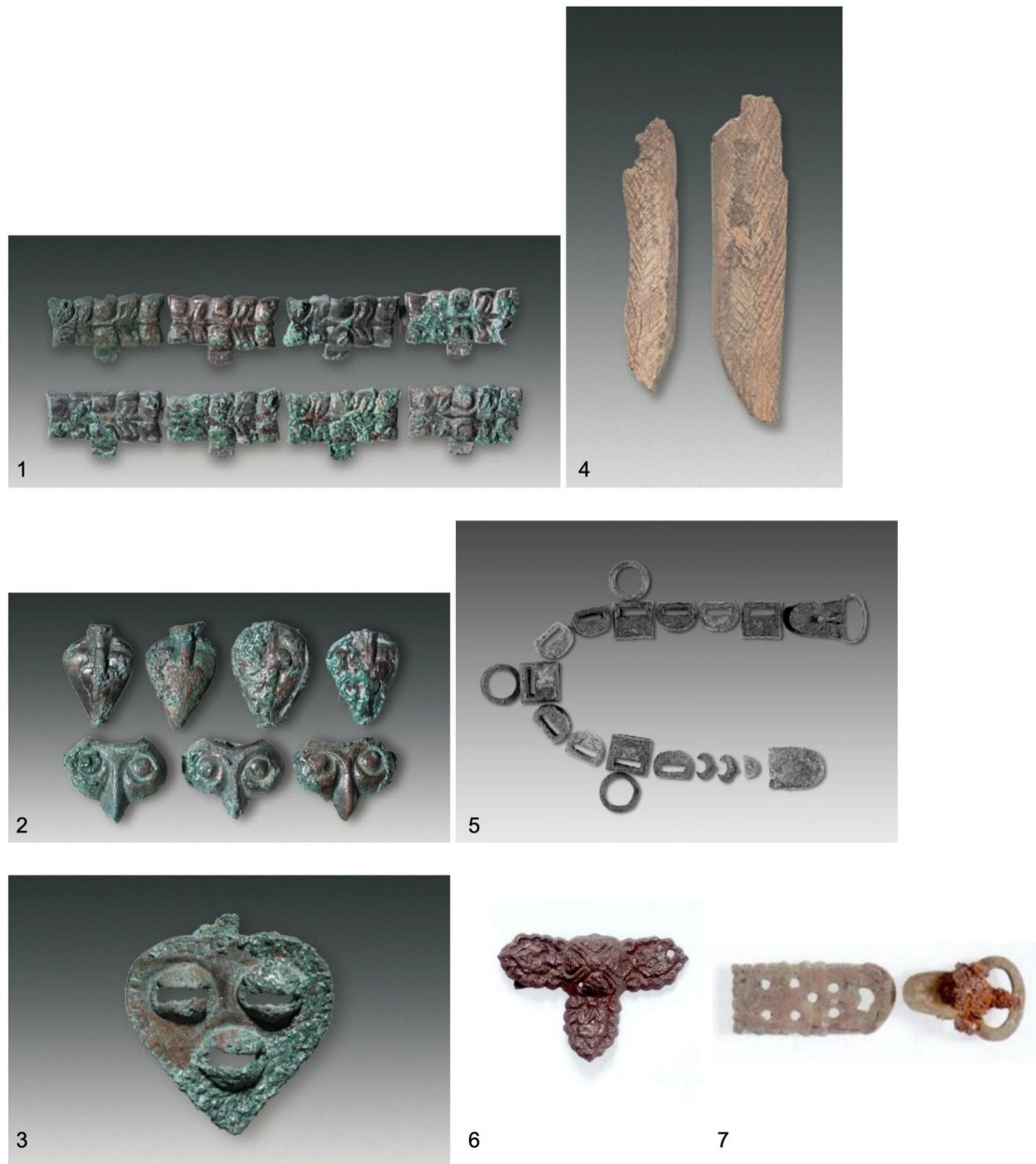

**Fig 15.** 1–3: bronze ornaments from M21 and 4: bone bow grip from M66 of Ningjiahe Reservoir Cemetery [49]. 5: bronze belt parts from M43 of the site of Gangou [51]. 6 and 7: bronze ornaments from M16 of Xigou Cemetery [48]. Scale not provided in original publications.

In general, vessels, especially ceramic ones, are rare in Turkic burials. More vessels have been uncovered in Kyrgyz burials [15] but they are also rare.

The only items of Chinese origin from Tunnug 1 are a bronze mirror and a Tang *kaiyuan-tongbao* coin. From the neighboring Altai-Sayan region and Mongolia, a total of 17 coins and 19 mirrors from a corpus of 450 graves have been reported. In addition, silk fragments were discovered in 50 graves; in 2 complexes, lacquerware was found [55]. Most of these items are

dated to the period of the Second Turk empire, between the second half of seventh and the first half of the eighth century.

No objects with runic inscriptions were discovered at Tunnug 1.

The relatively meager material finds from Tunnug 1 unfortunately do not provide a reliable index of the sociocultural makeup of the occupants, nor age and gender differences, but it can be observed that the burials contain similar hallmarks of military elitism and equestrian prestige associated with the archetypical Turkic burials.

## Conclusion

Research of Turkic archaeological culture has been heavily centered on the most distinct objects of burial and commemoration, which are principally evidence demonstrative of the prestige of horsemanship and military death. Although studies of these prominent finds are fruitful, they have inadvertently contributed to the segregation of archaeological data by region and therefore an over-reliance on establishing categories of analysis based on the area where diagnostic material finds are most concentrated, the Russian Altai. Our case study of Tunnug 1 in the contiguous Uyuk Valley showcases the limitations of such an approach in illustrating long-term trends of site use and regional funerary tradition particularly where archetypal material evidence is scant. The Tunnug 1 complex comprises a variety of burial practices that clustered over time around the locale of a monumental early Iron Age (9th c. BCE) burial mound. Artifacts characteristic of Turkic funerary culture—weaponry, tack, horse burial, anthropomorphic stela—are found across several groups of stone mounds although many were uncovered in contexts that had been heavily disturbed by solifluction or anthropogenic agents. Finding both inhumations and cremations with very similar radiocarbon dates and types of items, a temporal proximity and ritual polymorphism best explains their coexistence. Elsewhere in Tuva Republic, sites of small and mid-sized necropoleis contain varied forms of funerary architecture illustrative of different ethnic or group affiliations.

We compared the results from Tunnug 1 to the pattern of Turkic funerary customs in Mongolia, Xinjiang, and the Russian Altai. We argued that cross-regional analysis provides an integrative view of the archaeological landscape of the Turks that also benefits the characterization of coeval autochthonous or allochthonous funerary traditions. The permutations of funerary customs are likely a result of the admixture of cultures brought about by the tumultuous politics of inter-tribal coalitions. Polymorphism in burial practice—from the use of the *ogradka* memorial enclosures, the installation of *balbal* stone posts and anthropomorphic stelae, to the interment of horse and battle paraphernalia in burials—likely arose from military itinerancy and the *ad hoc* nature of funerals in traveling parties. This probably also explains why a funerary code of practice honoring battle sacrifices and equestrian skills in burial was widely and uniformly adopted, except in later burials in central Mongolia and Xinjiang where Turkic persons were interred in tombs modeled on Tang Chinese funerary architecture and furnished with murals and clay figurines.

Rather than viewing Turkic funerary customs as a discrete cultural trait that correlates ethnically with the Turks, a more comprehensive way of synthesizing the wide array of burial evidence across Inner Asia is to approach them as composites of diverse lines of burial heritage that at one time or another represented Turkic- or non-Turkic-speaking peoples of different north and central Asian descents. As Ecsedy [19: 284] remarked, "The burial customs thus represent only one and not at all exclusive, even if important cultural component that could at least contribute to historical processes of ethnic character"; in the case of the Turks, their archaeological record is congruous with their historically polymorphic character, and can be better understood as such.

## Acknowledgments

We would like to thank Alexey Kasparov (IHMC RAS) for the paleozoological definitions, and Zhang Yang for research assistance.

## Author Contributions

**Conceptualization:** Annie Chan.

**Data curation:** Timur Sadykov, Jegor Blochin, Irka Hajdas, Gino Caspari.

**Formal analysis:** Annie Chan, Timur Sadykov, Irka Hajdas, Gino Caspari.

**Funding acquisition:** Timur Sadykov, Jegor Blochin, Gino Caspari.

**Investigation:** Annie Chan, Timur Sadykov, Jegor Blochin, Gino Caspari.

**Methodology:** Annie Chan, Timur Sadykov, Jegor Blochin, Gino Caspari.

**Project administration:** Gino Caspari.

**Resources:** Annie Chan, Gino Caspari.

**Software:** Jegor Blochin, Gino Caspari.

**Supervision:** Gino Caspari.

**Validation:** Annie Chan.

**Visualization:** Timur Sadykov, Jegor Blochin, Gino Caspari.

**Writing – original draft:** Annie Chan, Timur Sadykov, Gino Caspari.

**Writing – review & editing:** Annie Chan, Timur Sadykov, Gino Caspari.

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
