## [Decision Letter · Decision Letter 0]

4 May 2022

PONE-D-22-05228The polymorphism and heritage of funerary practice of the medieval Turks in light of new findings from Tuva RepublicPLOS ONE

Dear Dr. Caspari,

Thank you for submitting your manuscript to PLOS ONE. After careful consideration, we feel that it has merit but does not fully meet PLOS ONE’s publication criteria as it currently stands. Therefore, we invite you to submit a revised version of the manuscript that addresses the points raised during the review process.

ACADEMIC EDITOR: Please insert comments here and delete this placeholder text when finished. Be sure to:Both reviewers agree (and I agree with them) that this is a very interesting and important paper that discusses Turkic archaeological heritage from the little known (in wider circles) region of Tuva. The paper brings a new set of radiocarbon dates as well as new material (from one site) and places it into a wider context.Although the paper has many qualities both reviewers spotted a few inconsistencies and provide detailed suggestions and comments on how to resolve certain issues. Reviewer 1 suggests to pay a closer attention to maps and illustrations to make them more clear and understandable as well as to be more consistent in the use of certain terms throughout the manuscript. Reviewer 2 also suggested some changes to the illustrations, but also provides a list of references to be used to strengthen the paper. This reviewer also suggests to add some technical details to the text such as soil/layer depths, dimensions of certain artefacts and so on. And finally, Reviewer 2 raises an important point that we cannot track the exact origin of the Turkic as an ethnic group and that the assumption that the Turks originated from the Altai has not yet been confirmed in archaeological materials, i.e. there is no such phenomenon as the "culture of the Turks" – please, pay close attention to this issue in your discussion.Please ensure that your decision is justified on PLOS ONE’s publication criteria and not, for example, on novelty or perceived impact.

We look forward to receiving your revised manuscript.

Kind regards,

Mario Novak

Academic Editor

PLOS ONE

Journal Requirements:

“The excavation project is conducted with the financial and logistical help of the Russian Geographical Society (N 53/04/2019) and financial support of Dr. F. Paulsen, the Society for the Exploration of EurAsia, and the Russian Ministry of Culture (project No. 656-01-1-41/12-18). Post-excavation works of T. S. and J. B. were carried out within the framework of the Programs of Fundamental Scientific Research of the Russian Academy of Sciences, State Assignments No. 0160-2020-0002 and 0184-2019-0011 respectively. The funders had no role in study design, data collection and analysis, decision to publish, or preparation of the manuscript. Thanks to Alexey Kasparov (IHMC RAS) for the paleozoological definitions, and Zhang Yang for research assistance.”

“The funders had no role in study design, data collection and analysis, decision to publish, or preparation of the manuscript. The excavation project is conducted with the financial and logistical help of the Russian Geographical Society (N 53/04/2019) and financial support of Dr. F. Paulsen, the Society for the Exploration of EurAsia, and the Russian Ministry of Culture (project No. 656-01-1-41/12-18). Post-excavation works of T. S. and J. B. were carried out within the framework of the Programs of Fundamental Scientific Research of the Russian Academy of Sciences, State Assignments No. 0160-2020-0002 and 0184-2019-0011 respectively. G. C. was funded by the Swiss National Science Foundation, grant number P400PG_190982.”

6. We note that you have referenced (Kilunovskaya et al. 2015, Kilunovskaya et al. 2017,Sadykov 2017b, Mongush and Mongush 2020) which has currently not yet been accepted for publication. Please remove this from your References and amend this to state in the body of your manuscript: (ie “Bewick et al. [Unpublished]”) as detailed online in our guide for authors

Reviewers' comments:

Reviewer's Responses to Questions

**Comments to the Author**

1. Is the manuscript technically sound, and do the data support the conclusions?

Reviewer #1: Yes

Reviewer #2: Yes

2. Has the statistical analysis been performed appropriately and rigorously? 

Reviewer #1: N/A

Reviewer #2: N/A

3. Have the authors made all data underlying the findings in their manuscript fully available?

Reviewer #1: Yes

Reviewer #2: Yes

4. Is the manuscript presented in an intelligible fashion and written in standard English?

Reviewer #1: Yes

Reviewer #2: Yes

5. Review Comments to the Author

Reviewer #1: Summury

In this interesting manuscript, Gino Сaspari and colleagues present an engaging study of early medieval materials from an important and little-explored area of the archaeological record (and one whose pertinent literature is often not shared in English). Although initially the research is based on the materials of only one site, in fact a kind of case study, the authors tried to bring the study to a new high level. The manuscript provides a detailed analysis of the Turkic antiquities of the neighboring regions of Inner Asia - Altai, Mongolia and Xinjiang. This allows the authors to consider the problem on a broader background and is important for historical interpretation.

The idea of polymorphism of Turkic burial customs is not new. The authors are right, noting the fact that the information of Chinese written sources does not match confirmation in the archeological data. This is especially true for cremations. The problem of belonging the burials of a man with a horse to specific Turkic ethnic groups is not fully understood. Recently, new, even exotic hypotheses have appeared in the literature, for example, about the belonging of all such burials in the Russian Altai to the Karluks (Kubarev, 2021). New archaeological data and a thorough analysis of various versions of translations of Chinese Dynastic Сhronicles are the keys to solving this problem.

I will especially note the series of radiocarbon dates that the authors publish. For this time and region, such data are rare due to objective reasons. Therefore, these data are of great importance for studying not only the Turkic burials in Tuva, but also in the neighboring regions of Central Asia and Southern Siberia.

I hope that the submitted manuscript will be of interest to a wide international academic audience and can be recommended for publication after a light revision.

Miscellaneous comments and suggestions

Keywords: it may be better to replace "mounted combat" with "horsemanship"

It is necessary to use the same terms throughout the text. For example: "Khanate" (lines 27, 28, 38, 74, 596 etc.) and "Khaganate" (line 173, 563 etc.), "Kok Turks" (line 114), "Gokturks" (line 122) and "Güktürks" (line 172), "Turkic" (as usual) but "Türkic period" (line 283, 292), "Altai" (that's right) and «Altay» (line 529).

(Lines 331-338) The authors explain the use of the terms "structures" and "objects". I think it is necessary to unify and choose one, because the term "structure" is used in the text, and "object" is used in the figures. This creates confusion.

All the titles of the publications in Russian in the text are given in transliteration, except for one book by Dmitry Savinov (line 87) given in Cyrillic. I recommend to follow one style.

The phrase "...gold and metal jewelry" (line 150) is not quite clear. I understand that this is supported by the reference, but it's better to rephrase.

The map in Fig. 1 shows the distribution of excavated Turkic burials in Inner Asia very well, but it may be better to add geographical landmarks so that it becomes more informative.

There are no symbols in Fig. 13. What do the triangles on the plan mean? Ceramics? If yes, add coments to the caption please.

What does the signature in Fig. 7 "ob.27 may not be included" mean?

It is necessary to check the scale for all items in Fig. 8. I have doubts about the size ratios of some plaques and buckles. In addition, it was better to show some items (plaques from object 7) horizontally, as they were usually attached to a belt.

If the limit of figures allows (I’m understand it is always problem), you need to add a map with the location of the Tunnug, and not be limited to a reference to own early publication. But it remains on the decision of the authors.

«The dog would have been about the size of a fox terrier or a large husky» (line 463). I believe that such a comparison with specific breeds of dogs is not correct.

Table 1. What software was used to calibrate radiocarbon dates?

Once again, this important study by an international team of researchers can be recommended after minor revision and hopefully will be useful for a wide academic audience.

Reviewer #2: This is an interesting study; the authors have collected new archaeological evidence of Turkic funerary customs and, more generally Turkic cultural heritage in Central Asia. The paper is well written and structured. The manuscript comes as a result of promising field research and excavations in the Valley of the Kings in Tuva. The article provides an exhaustive list of literature, taking into account the latest research. The paper can be relevant to a broader readership, such as archaeologists, cultural anthropologists, ethnographers, and historians. The archaeological context is well presented for Central Asia in the Early Middle Ages. The authors discuss a broad range of cultural analogies published by other authors. The originality and interpretative ideas of the authors are based on new radiocarbon data, which confirms the legitimacy of correlating described monuments with the time of the Turkic Khaganate and the era of the early Middle Ages.

However, in my opinion, the paper has some shortcomings in regards to some formal aspects and text, and I feel that collected evidence has not been utilized to its full extent. Below the authors and the editor can find the remarks.

-line 555-556: while discussing the meaning of fences in medieval burial practices of the region, I strongly support the idea of the memorial purpose of fences – the so-called “ogradki”; since there are no arguments in favor of the fact that these are cenotaphs, and I agree with the arguments of the authors against the fact that these are cenotaphs.

- Figs 5, 8: In the article, we do not see an analysis of the typology of the bits and stirrups found on the monument. Also, the latest research on this problem is not cited (Seregin 2017; 2018).

-When analyzing the culture of the early Middle Ages, it would be useful to cite the work of Dosymbaeva (Dosymbaeva 2006).

-line 665: The problem is that we cannot track the exact origin of the Turkic as an ethnic group. The information/assumption that the Turks originated from the Altai has not yet been confirmed in archaeological materials. There is no such phenomenon as the "culture of the Turks" in the vastness of central Eurasia. It can also be interpreted as many different cultures. Whether different cultures can be called polymorphism of "Turkic culture" is an open question. Consider in the discussion.

-line 434, Fig.11: It should be noted that the monuments studied by the authors are not analyzed from the point of view of the spatial orientation of the graves and bodies interred. It is known, that the Turks had certain preferences in this respect: an eastern alignment of sacred objects, burials, dwellings. But the article does not consider the question of the monument's compliance with this cultural trait. Culturally based preferences (esp. E-W alignment) typical for Turks are evident in Fig.11, for example. Consider updating the manuscript.

-lines 373-374: It would be useful to use Western examples of funerary "fences" - in the form of typologically similar ditches in the burial mounds of the Khazars (Kruglov 2018) and the fence of the Voznesensky complex on the Dnieper (Ambrose 1982). They are generated by the same Turkic cultural environment and reflect the same ideas as the "fences" in Central Asia. In the Voznesensky memorial complex, we see traces of post-funeral rituals, when burned stirrups and weapons were placed in a pit after regular post-funeral feasts for several years.

-Fig.1. Formal correction: could you consider adding the names of the countries on the map.

-Fig.3. Formal correction: could you consider adding the depths (picture B for example)? Some more technical information about these soil covers (type etc.) would be useful e.g. cracking caused by…, consider adding a citation to some soil studies if necessary.

-Fig. 4. Formal suggestion: illustration is showing Tunnug 1. Medieval artifacts. Iron horse bits- can you add some more information, what elements is it exactly, what material is it made of, state of preservation (partially/full preserved, etc.), museum collection number? The same applies to Fig. 4-6; very vague description, and it can be improved. This is rather an en masse material, but in Fig.7 there is some strange note on a picture ‘maybe not include’- please remove it, or correct it differently.

- Fig. 8- Bronze finds from Objects 6 and 7- can you describe or name these items? function etc. Perhaps it could be a good idea to provide in descriptions some information about typologies of arrowheads by citation of appropriate published works. Fig. 8- scale in cm. Unify the fonts and scales in all illustrations. In Figs. 7-8 treat every item separately labeling them with numbers or letters, arrowhead 1, 2,3 – feature number 6, arrowhead 4,5,6,7 -feature number 27, etc. Fig. 9- maybe adding the author of the photo: photo by XY, that applies to all photos and drawings in the manuscript, drawn by/photo by, or if taken from a book, drawing after citation;

-Fig.14: formal suggestion: in the description please add dimensions of the stele, the stone material it was made of, and optionally weight. It would be useful to mark pictures a,b,c; the stele had been decorated, and there is some almond shape in the upper part, can you comment on that? is this perhaps anthropomorphic, or if not, add some comment regarding potential decorations. Add the author of these photos.

-line 575,587-590- change of the font? there is something wrong with the font format here; please also check the citation format: example: Linghu 1971: 910 (vol. 50, biography no. 42, “Tujue”, or Wang 1935: 1784 (vol. 100); these are Chinese works but the format of citation should match modern standards somehow; the dating of Chinese Dynasties if mentioned also should have AD/BC.

-Fig.15 I would like to suggest inserting item no. 4 in a horizontal position so the illustration keeps a more regular shape.

-lines 613-614 ‘Russian archaeologists such as V.V. Gorbunov, G.V. Kubarev, V.D. Kubarev, N.N. Seregin, A.A.Tishkin, have published compendiums of these artifacts… etc. Please add all mentioned works as citations by the end of the sentence.

But, despite these remarks, the article can be published without any doubt. The authors' conclusions are based on a large amount of factual material and correct theoretical generalizations. Below I recommend some more works which the authors may find useful for their manuscript in revison process:

Ambrose. A.K. About the Voznesensky complex of the VIII century on the Dnieper — a question of interpretation // Antiquities of the epoch of the Great Migration of peoples V to VIII centuries. Moscow: 1982. pp. 204-222. In Russian

Dosymbaeva A.M. Western Turkic Khaganate. Cultural heritage of the Kazakh steppe. Almaty, 2006. In Russian

Kruglov, E. V. FORMATION OF THE SOURCE BASE OF THE SOKOLOVSKY BALKATIP MONUMENTS // Nizhnevolzhsky Archaeological Bulletin. – 2018. – Vol. 17. – No. 1. – pp. 144-159. – DOI 10.15688/nav.jvolsu.2018.1.7. In Russian

Seregin, N. N. Stirrups from the burial complexes of the early medieval Turks in Mongolia // Peoples and religions of Eurasia. – 2017. – № 3-4(12-13). – Pp. 9-23.

Seregin, N. N. Horse bits and cheeks from early medieval Turkic burials in Mongolia // Izvestiya Altaiskogo gosudarstvennogo universiteta. – 2018. – № 2(100). – Pp. 174-181. – DOI 10.14258/izvasu(2018)2-29. In Russian

In Russian:

Серегин, Н. Н. Стремена из погребальных комплексов раннесредневековых тюрок Монголии // Народы и религии Евразии. – 2017. – № 3-4(12-13). – С. 9-23.

Серегин, Н. Н. Удила и псалии из погребальных комплексов раннесредневековых тюрок Монголии // Известия Алтайского государственного университета. – 2018. – № 2(100). – С. 174-181. – DOI 10.14258/izvasu(2018)2-29.

Амброз А.К. О Вознесенском комплексе VIII в. на Днепре — вопрос интерпретации // Древности эпохи великого переселения народов V-VIII вв. М.: 1982. С. 204-222.

Досымбаева А.М. Западный Тюркский каганат. Культурное наследие казахской степи. Алматы, 2006.

Круглов, Е. В. Формирование источниковой базы памятников типа Соколовской балки. Шилов В.П.: курганный могильник Ордынский бугор // Нижневолжский археологический вестник. – 2018. – Т. 17. – № 1. – С. 144-159. – DOI 10.15688/nav.jvolsu.2018.1.7.

6. PLOS authors have the option to publish the peer review history of their article (what does this mean?). If published, this will include your full peer review and any attached files.

Reviewer #1: **Yes: **Alexey Fribus

Reviewer #2: **Yes: **Dalia Pokutta, Ph.D., Department of Archaeology University of Stockholm (ARL) Sweden; Department of Archaeology and Museology Masaryk University, Czech Republic

---

## [Author Response · Author response to Decision Letter 0]

18 Jul 2022

see "Response to Reviewers" file

18.07.2022 Addition:

The second resubmission was sent back by the manuscript check. We address the raised points below:

1. It is important that you include a cover letter with your manuscript. Please ensure that this letter is addressed specifically to PLoS ONE. Please also include

* why this manuscript is suitable for publication in PLoS ONE.

* how does your paper provide a worthwhile addition to the scientific literature?

* how does your paper relate to previously published work?

* which types of scientists do you believe will be most interested in your study?

I REPEAT: We had included a cover letter – as always. It was addressed to PLOS ONE as can be seen above. As this is a resubmission of a minor revision, we believe that the editor has already judged the manuscript being potentially suitable for and interesting to the audience of PLOS ONE. 

For the manuscript checker’s convenience we add the statement again:

In this work, we contribute new data and an interpretative framework for the variability of Turkic funerary rituals in the Eastern Steppe. The medieval Turks of the eastern Asian steppe are known for funerary finds exalting horsemanship and military heroism that thrived on intertribal warfare. Existing bodies of research on various categories of objects — which include architecture, stelae, grave goods and inhumations — are in depth but highly regionalized. As a result, our understanding of the archaeological culture of the Turks on a spatio-temporal scale commensurate with territorial shifts in their political diminion from the First Turkic Khanate through the Second Turkic Khanate (mid-6th to mid-8th centuries) remains disjunct. This paper addresses this problem of disparate data. We present a synthesis of Turkic archaeological research spanning Mongolia, southern Siberia, and Xinjiang in view of results of the excavation of medieval burials at Tunnug 1 in Tuva Republic— where Turkic remains are dispersed and not easily distinguishable from other funerary cultures of connecting time periods. We argue that Turkic funerary culture can be better characterized as polymorphic, which describes the presence of different regional amalgams of burial traditions. The horse-and-human burials and commemorative ogradka known to be quintessentially Turkic are but one of the more dominant amalgams. This pattern of differential practices is congruent with the history of the Turks as peoples of different lineages and political groupings, rather than people of a unitary culture.

2. Please upload a Response to Reviewers letter which should include a point by point response to each of the points made by the Editor and / or Reviewers. (This should be uploaded as a 'Response to Reviewers' file type.) 

As in the previous submission you can find the Response to Reviewers point by point below. 

3. Thank you for including your ethics statement on the online submission form: "Field research was conducted under licenses No. 0434–2018 and No. 0590–2019 issued to T.S. from the Institute for the History of Material Culture, Russian Academy of Sciences. The permits were issued by the Russian Ministry of Culture." To help ensure that the wording of your manuscript is suitable for publication, would you please also add this statement at the beginning of the Methods section of your manuscript file.

We added the statement to the methods section. Thank you. 

4. We note that you have referenced (Kilunovskaya et al. 2015, Kilunovskaya et al. 2017,Sadykov 2017b, Mongush and Mongush 2020) which has currently not yet been accepted for publication. Please remove this from your References and amend this to state in the body of your manuscript: (ie “Bewick et al. [Unpublished]”) as detailed online in our guide for authors

I REPEAT: This has been noted before and it is likely a problem with an automated check that does not read the citations correctly. All sources have been published.

5. Thank you for your response regarding the potential copyright of your Figure. We apologize for any delays incurred as a result of our checks for copyright.

Regarding Figure 1, please respond to the following:

a) As you note you have used ArcMap for the creation of this figure, please update your Figure caption to include attribution to ESRI. Upon resubmission, this should be good to proceed.

Added ESRI attribution.

---

## [Decision Letter · Decision Letter 1]

8 Aug 2022

PONE-D-22-05228R1The polymorphism and tradition of funerary practices of medieval Turks of Central Asia in light of new findings from Tuva RepublicPLOS ONE

Dear Dr. Caspari,

Thank you for submitting your manuscript to PLOS ONE. After careful consideration, we feel that it has merit but does not fully meet PLOS ONE’s publication criteria as it currently stands. Therefore, we invite you to submit a revised version of the manuscript that addresses the points raised during the review process.

 The manuscript is almost ready to be accepted. Please, address two minor suggestions proposed by the reviewers and return the manuscript back. Once you resolve these two issues the paper will be ready two be accepted. 

We look forward to receiving your revised manuscript.

Kind regards,

Mario Novak

Academic Editor

PLOS ONE

Journal Requirements:

Additional Editor Comments:

Both reviewers agree that all issues have been properly addressed and that the manuscript can be accepted for publication. However, before I accept it, please address two minor suggestions by both reviewers.

Reviewers' comments:

Reviewer's Responses to Questions

**Comments to the Author**

1. If the authors have adequately addressed your comments raised in a previous round of review and you feel that this manuscript is now acceptable for publication, you may indicate that here to bypass the “Comments to the Author” section, enter your conflict of interest statement in the “Confidential to Editor” section, and submit your "Accept" recommendation.

Reviewer #1: (No Response)

Reviewer #2: All comments have been addressed

2. Is the manuscript technically sound, and do the data support the conclusions?

Reviewer #1: Yes

Reviewer #2: Yes

3. Has the statistical analysis been performed appropriately and rigorously? 

Reviewer #1: N/A

Reviewer #2: N/A

4. Have the authors made all data underlying the findings in their manuscript fully available?

Reviewer #1: Yes

Reviewer #2: Yes

5. Is the manuscript presented in an intelligible fashion and written in standard English?

Reviewer #1: Yes

Reviewer #2: Yes

6. Review Comments to the Author

Reviewer #1: I believe that the revised article can be published. Please correct the typo: page 4 line 2: Seregin and Matrenin (not Maternin)

Reviewer #2: (No Response)

7. PLOS authors have the option to publish the peer review history of their article (what does this mean?). If published, this will include your full peer review and any attached files.

Reviewer #1: **Yes: **Alexey Fribus

Reviewer #2: **Yes: **Dalia Pokutta

---

## [Author Response · Author response to Decision Letter 1]

14 Aug 2022

14.08.2022

Minor typos corrected. Ready for acceptance.

---

## [Editor Report · Decision Letter 2]

31 Aug 2022

The polymorphism and tradition of funerary practices of medieval Turks in light of new findings from Tuva Republic

PONE-D-22-05228R2

Dear Dr. Caspari,

We’re pleased to inform you that your manuscript has been judged scientifically suitable for publication and will be formally accepted for publication once it meets all outstanding technical requirements.

Kind regards,

Mario Novak

Academic Editor

PLOS ONE
---

## [Editor Report · Acceptance letter]

6 Sep 2022

PONE-D-22-05228R2 

The polymorphism and tradition of funerary practices of medieval Turks in light of new findings from Tuva Republic 

Dear Dr. Caspari:

I'm pleased to inform you that your manuscript has been deemed suitable for publication in PLOS ONE. Congratulations! Your manuscript is now with our production department. 

Kind regards, 

on behalf of

Dr. Mario Novak 

Academic Editor

PLOS ONE